# BOOSTING OFFLINE MULTI-OBJECTIVE REINFORCEMENT LEARNING VIA PREFERENCE CONDITIONED DIFFUSION MODELS

## ABSTRACT

Multi-objective reinforcement learning (MORL) addresses sequential decision-making problems with multiple objectives by learning policies optimized for diverse preferences. While traditional methods necessitate costly online interaction with the environment, recent approaches leverage static datasets containing pre-collected trajectories, making offline MORL the preferred choice for real-world applications. However, existing offline MORL techniques suffer from limited expressiveness and poor generalization on out-of-distribution (OOD) preferences. To overcome these limitations, we propose **Diff**usion-based **M**ulti-**O**bjective **R**einforcement **L**earning (DIFFMORL), a generalizable diffusion-based planning framework for MORL. Leveraging the strong expressiveness and generation capability of diffusion models, DIFFMORL further boosts its generalization through offline data *mixup*, which mitigates the memorization phenomenon and facilitates feature learning by data augmentation. By training on the augmented data, DIFFMORL is able to condition on a given preference, whether in-distribution or OOD, to plan the desired trajectory and extract the corresponding action. Experiments conducted on the D4MORL benchmark demonstrate that DIFFMORL achieves state-of-the-art results across nearly all tasks. Notably, it surpasses the best baseline on most tasks, underscoring its remarkable generalization ability in offline MORL scenarios.

## 1 INTRODUCTION

Reinforcement learning (RL) (Wang et al., 2024) empowers an agent to learn to achieve a specific objective through interactions with the environment, and has made exciting progress in various real-world problems like autonomous driving (Kiran et al., 2022), robotic control (Singh et al., 2022), and healthcare (Yu et al., 2023), etc. While the classic RL framework focuses on optimizing a single objective through the maximization of a scalar return, multi-objective RL (MORL) (Roijers et al., 2013; Liu et al., 2014) endeavors to optimize multiple competing objectives associated with a vector-valued reward. The majority of MORL approaches (Abels et al., 2019; Xu et al., 2020; Yang et al., 2019; Basaklar et al., 2023; Hung et al., 2023; Lin et al., 2024a) learn a set of policies optimized for diverse preferences over the objectives, allowing for the selection of the most suitable policy based on user preferences during deployment. For instance, a MORL healthcare agent can recommend an appropriate treatment plan based on different patient preferences and medical requirements. However, these approaches adopt an online learning paradigm, entailing extensive interactions with the environment to effectively learn a wide range of preferences. It poses practical challenges in real-world problems where data collection is costly and potentially hazardous.

Learning from static datasets with pre-collected trajectories corresponding to different preferences, offline MORL methods emerge as the preferred choice to solve this issue. For instance, PEDI (Wu et al., 2021) transforms the original offline multi-objective problem into a primal-dual formulation and solves it via dual gradient ascent. Another method, PEDA (Zhu et al., 2023a), extends return-conditioned methods including Decision Transformer (DT) (Chen et al., 2021a), RvS (Emmons et al., 2022), and primitive diffusion (Yuan et al., 2024) with two return normalizations to the multi-objective setting. Some works recently develop policy-regularized methods to improve the learning efficiency of offline MORL (Lin et al., 2024b). Meanwhile, researchers also develop offline MORL benchmarks, including D4MORL (Zhu et al., 2023a), which evaluates the Pareto-efficiency of the agents via a

wide range of tasks, and MOSB (Lin et al., 2024b), which focuses on assessing the feasibility of utilizing single objective datasets such as D4RL (Fu et al., 2020). These advancements have propelled offline MORL to take a significant step forward in addressing multi-objective real-world problems.

However, current offline MORL methods suffer from limited expressiveness and struggle to accurately model the diverse optimal policies that correspond to a wide range of preferences, leading to the suboptimality of the approximated Pareto front. Additionally, these methods do not explicitly consider the limited preference coverage of offline datasets, but rather learn from the limited datasets directly. Consequently, these methods perform well only on the preferences covered within the dataset but generalize poorly on out-of-distribution (OOD) preferences. Thus, a question arises: *can we develop an offline multi-object reinforcement learning approach that strengthens the agent's generalization ability using only limited offline data?*

For the mentioned issue, we propose **Diff**usion-based **M**ulti-**O**bjective **R**einforcement **L**earning (DIFFMORL), a strong and generalizable diffusion-based planning framework for offline MORL. It leverages the well-established expressiveness and generation capability of diffusion models (Yang et al., 2023) to model the policies. Furthermore, to enhance generalization to OOD preferences, instead of conservatively selecting in-distribution policies with the closest preference, i.e., the memorization phenomenon, DIFFMORL applies the widely used *mixup* technique (Zhang et al., 2018; Cao et al., 2022; Jin et al., 2024) to synthesize pseudo-trajectories and augment the learning process. Experiments conducted on the D4MORL (Zhu et al., 2023a) benchmark demonstrate that DIFFMORL achieves state-of-the-art results across nearly all multi-objective MuJoCo-based (Todorov et al., 2012) continuous control tasks. Notably, DIFFMORL surpasses the best baseline on most of tasks in terms of Return Mismatch, a metric to measure the performance on OOD preferences, underscoring its remarkable generalization ability in offline MORL scenarios.

## 2 RELATED WORK

**Offline Multi-Objective Reinforcement Learning (MORL)** MORL extends the classic RL framework from a single optimization objective to multi-objective settings (Hayes et al., 2022), making it well-suited for real-world problems such as transportation (Ren et al., 2021) and hyperparameter tuning (Chen et al., 2021b). The majority of MORL approaches aim to learn a set of policies that approximates the Pareto front in an online paradigm. For instance, PG-MORL (Xu et al., 2020) updates a policy population using an evolutionary algorithm, while approaches like Envelope (Yang et al., 2019), PD-MORL (Basaklar et al., 2023), and Q-Pensieve (Hung et al., 2023) train a single preference-conditioned network with different Bellman update strategies, which may be impractical in critical domains such as healthcare and autonomous driving, accelerating the focus on the offline MORL setting. Offline MORL adopts an offline learning paradigm, deriving policies from static datasets. PEDI (Wu et al., 2021) transforms the offline multi-objective problem into a primal-dual formulation solved via dual gradient ascent, while PEDA (Zhu et al., 2023a) extends return-conditioned sequential modeling methods to the multi-objective setting. Policy-regularized methods have also been applied to address preference-inconsistent demonstrations (Lin et al., 2024b). Very recently, MODULI (Yuan et al., 2024), using a preference-conditioned diffusion model as a planner to generate trajectories aligned with various preferences, shows potential for improving offline MORL efficiency in ideal settings and exhibits generalization ability in out-of-distribution scenarios. Researchers have developed offline MORL benchmarks, such as D4MORL (Zhu et al., 2023a), which evaluates agents' Pareto-efficiency across a wide range of tasks, and MOSB (Lin et al., 2024b), which assesses the feasibility of using single-objective datasets like D4RL (Fu et al., 2020).

**Diffusion Models in RL** Diffusion models have emerged as a powerful generative modeling framework in machine learning. These models employ a Markov chain to gradually add noise to the data, followed by a learned denoising process to generate new samples (Yang et al., 2023). Their effectiveness has been demonstrated across a wide range of domains, including computer vision (Croitoru et al., 2023), video generation (Ho et al., 2022), and text-to-image synthesis (Qin et al., 2024), among others. In reinforcement learning (RL), diffusion models have initially been applied to planning tasks, exemplified by methods such as Diffuser (Janner et al., 2022) and Decision Diffuser (Ajay et al., 2023). More recent work has explored the use of diffusion models for policy parameterization, where they generate action sequences (Lin et al., 2024b; Wang et al., 2022), and for data augmentation, where they synthesize new data (Lu et al., 2024; Yang & Xu, 2024). While

diffusion models have shown success in single-agent settings, approaches like MADiff (Zhu et al., 2023b) and EAQ (Oh et al., 2024) have extended their application to multi-agent environments, significantly improving multi-agent coordination and learning efficiency. Diffusion models have also been applied in robotics and large language models (LLMs) (Zhu et al., 2023c), showcasing their high expressive power and problem-solving capabilities across various problem settings.

**Mixup Augmentations**   Mixup is a data augmentation with the core idea being to generate new synthetic training samples by linearly interpolating between two images and their corresponding labels (Zhang et al., 2018; Jin et al., 2024). By encouraging the model to make smooth predictions over these interpolated data points, mixup has been proven highly effective in reducing overfitting and improving generalization, particularly when dealing with limited or noisy datasets. It has shown great potential in areas such as computer vision (Xu et al., 2023), point cloud processing (Chen et al., 2020), and natural language processing (NLP) (Sun et al., 2020). In reinforcement learning, mixup has also been applied to improve generalization. For instance, Mixreg (Wang et al., 2020) trains agents by mixing observations from different training environments and enforces linearity constraints on both the interpolated observations and associated rewards, while MixRL (Hwang & Whang, 2021), a data augmentation meta-learning framework for regression, identifies the optimal number of nearest neighbors to mix for each sample to improve model performance using a small validation set. Additionally, K-mixup incorporates mixup into reinforcement learning by learning a Koopman invariant subspace, a method commonly used for classification tasks (Jang et al., 2023). Other works, such as (Ajay et al., 2023), employ mixup to train classifiers that validate the generalization of diffusion models.

## 3 PRELIMINARIES

**Multi-Objective Markov Decision Process (MOMDP)**   We formulate the multi-objective sequential decision making problem as a Multi-Objective Markov Decision Process (MOMDP) with linear preferences (Wakuta, 1995), defined by the tuple $\langle \mathcal{S}, \mathcal{A}, \mathcal{P}, \mathcal{R}, \Omega, f, \gamma \rangle$, where $\mathcal{S}$ and $\mathcal{A}$ denote the state space and the action space. $\mathcal{P} : \mathcal{S} \times \mathcal{A} \to \Pr(\mathcal{S})$ is the transition function, $\mathcal{R} : \mathcal{S} \times \mathcal{A} \to \mathbb{R}^n$ is the vector-valued reward function and $n$ is the number of objectives. We also assume that there exists a preference space $\Omega \in \Pr(\mathbb{R}^n)$ and a linear utility function $f : \Omega \times \mathbb{R}^n \to \mathbb{R}$ that scalarize the reward vector $\boldsymbol{r}_t = \mathcal{R}(\boldsymbol{s}_t, \boldsymbol{a}_t)$ as $r_t = f(\boldsymbol{\omega}, \boldsymbol{r}_t) = \boldsymbol{\omega}^\top \boldsymbol{r}_t$, given preference $\boldsymbol{\omega} \in \Omega$. At timestep $t$, an agent with state $\boldsymbol{s}_t \in \mathcal{S}$ executes an action $\boldsymbol{a}_t \in \mathcal{A}$, and then transition to the next state $\boldsymbol{s}_{t+1}$ with probability $\mathcal{P}(\boldsymbol{s}_{t+1}|\boldsymbol{s}_t, \boldsymbol{a}_t)$, and receive a vector-valued reward $\boldsymbol{r}_t$. The vector-valued return is given by the discounted sum of reward vectors as $\boldsymbol{R} = \sum_t \gamma^t \boldsymbol{r}_t$. The expected vector-valued return for a policy $\pi(\boldsymbol{a}|\boldsymbol{s}, \boldsymbol{\omega})$ is $\boldsymbol{G}^\pi = \mathbb{E}_{\boldsymbol{s}_0, \boldsymbol{a}_t \sim \pi(\cdot|\boldsymbol{s}_t, \boldsymbol{\omega})}[\boldsymbol{R}]$, and the goal is to train a multi-objective policy $\pi$ that maximize the expected scalarized return $\boldsymbol{\omega}^\top \boldsymbol{G}^\pi, \forall \boldsymbol{\omega} \in \Omega$.

**Diffusion Probabilistic Models**   Diffusion models have two process, the forward process gradually adds noises to the clean samples $\boldsymbol{x}$ via a pre-scheduled diffusion function $q(\boldsymbol{x}_{k+1}|\boldsymbol{x}_k) := \mathcal{N}(\boldsymbol{x}_{k+1}|\sqrt{\alpha_k}\boldsymbol{x}_k, (1 - \alpha_k)\boldsymbol{I})$. On the contrary, the reverse process gradually removes noises from the noisy samples $\boldsymbol{x}_k$ via a learnable function $p_\theta(\boldsymbol{x}_{k-1}|\boldsymbol{x}_k) = \mathcal{N}(\boldsymbol{x}_{k-1}|\mu_\theta(\boldsymbol{x}_k, k), \Sigma_k)$, where $\mathcal{N}(\boldsymbol{x}|\mu, \Sigma)$ is a Gaussian distribution with mean vector $\mu$ and covariance matrix $\Sigma$, $\boldsymbol{x}_0 = \boldsymbol{x}$ is a sample, $\boldsymbol{x}_1, \dots, \boldsymbol{x}_K$ are noisy latent variables, $\alpha_k \in \mathbb{R}$ are coefficients that determine the variance schedule, and $K$ is the predefined maximal diffusion timestep. A sample $\boldsymbol{x}$ can be generated by running the reverse process to iteratively denoise a prior $\boldsymbol{x}_K \sim \mathcal{N}(\boldsymbol{0}, \boldsymbol{I})$ for $K$ steps. To efficiently train diffusion models to derive $p_\theta$, DDPM (Ho et al., 2020) runs the forward process and employs a neural network $\epsilon_\theta$ to predict the noises, i.e., minimizing the loss:

$$\mathcal{L}(\theta) = \mathbb{E}_{k, \boldsymbol{x}_0, \epsilon}\left[\|\epsilon - \epsilon_\theta(\boldsymbol{x}_k, k)\|^2\right], \qquad (1)$$

where $k$ is uniformly sampled from $\{1, \dots, K\}$, $\boldsymbol{x}_0$ is a sample, $\epsilon \sim \mathcal{N}(\boldsymbol{0}, \boldsymbol{I})$ is noise, $\boldsymbol{x}_k = \sqrt{\bar{\alpha}_k}\boldsymbol{x}_0 + \sqrt{1 - \bar{\alpha}_k}\epsilon$ is the noisy sample, and $\bar{\alpha}_k := \prod_{s=1}^k \alpha_s$. The reverse process $p_\theta$ is equivalent to noise prediction using $\epsilon_\theta$, as denoising is exactly removing predicted noises from noisy samples.

Conditional diffusion models are developed with posterior $p_\theta(\boldsymbol{x}_{k-1}|\boldsymbol{x}_k, \boldsymbol{y})$ that denoise with additional information $\boldsymbol{y}$, and the noises are predicted by the conditional network $\epsilon_\theta(\boldsymbol{x}_k, \boldsymbol{y}, k)$. These models are able to generate samples according to some attributes, flexibly synthesizing novel behaviors. Essentially, there is an equivalence between diffusion models and score matching,

which shows $\epsilon_\theta(\boldsymbol{x}_k, k) \propto \nabla_{\boldsymbol{x}_k} \log p(\boldsymbol{x}_k)$, i.e., the noise is proportional to gradient (score) of the data distribution. This relationship leads to a score-based conditioning trick of diffusion models. Classifier-free guidance is one implementation that learn a conditional $\epsilon_\theta(\boldsymbol{x}_k, \boldsymbol{y}, k)$ and an unconditional $\epsilon_\theta(\boldsymbol{x}_k, \varnothing, k)$ at the same time, where $\varnothing$ is a fixed dummy value. Then, the perturbed noise $\hat{\epsilon} = \epsilon_\theta(\boldsymbol{x}_k, \varnothing, k) + w[\epsilon_\theta(\boldsymbol{x}_k, \boldsymbol{y}, k) - \epsilon_\theta(\boldsymbol{x}_k, \varnothing, k)]$ is used for generation (Song et al., 2021).

## 4 METHOD

In this section, we present the detailed design of the proposed framework, DIFFMORL, for generalizable offline MORL. First, we formulate the problem of OOD preferences, and the trajectory generation process for task planning in Section 4.1. Next, in Section 4.2, we describe the training methodology for DIFFMORL, where we utilize the mixup technique to enhance generalization. Finally, we explain how to plan and execute MORL tasks using DIFFMORL in Section 4.3.

### 4.1 PROBLEM SETUP

**The Problem of OOD Preferences**   In real-world offline MORL tasks, the pre-collected dataset $\mathcal{D}$ may suffer from *incomplete* preference coverage, due to the property of tasks and behavior policies. For example, preferences that treat all objectives almost equally or unilaterally may be lacking in some scenarios (Figure 1). To capture this issue, we define the *preference-lacking region* as the union of sets $B(\boldsymbol{\omega}_{\text{ood}}, \epsilon) = \{\boldsymbol{\omega} \in \Omega \mid \|\boldsymbol{\omega} - \boldsymbol{\omega}_{\text{ood}}\|_1 \leq \epsilon\}$, for a series of $\epsilon \geq \epsilon_{\min}$ and $\omega_{\text{ood}}$, where $\epsilon_{\min}$ is a positive constant for ensuring the inevitability of the region. These preferences are termed out-of-distribution (OOD) preferences due to their absence from the dataset. Offline MORL algorithms that learn directly from such *incomplete* datasets may derive suboptimal policies when evaluated on OOD preferences, i.e., poor generalization. The following sections will provide an detailed approach to addressing this problem.

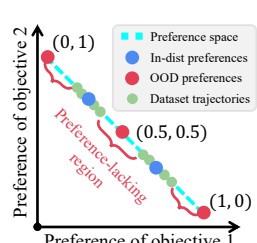

Figure 1: An example of the OOD preference problem.

**Trajectory Generation via Diffusion**   To capture the complex distribution of trajectories across a wide range of preferences and returns, we formulate the MORL planning problem as a conditional generation problem using a diffusion model:

$$\max_\theta \mathbb{E}_{\tau \sim \mathcal{D}}[\log p_\theta(\boldsymbol{x}(\tau)|\boldsymbol{y}(\tau))], \tag{2}$$

where $\mathcal{D}$ is a pre-collected offline MORL dataset containing trajectories of the form $\tau = \langle \boldsymbol{\omega}, \boldsymbol{s}_1, \boldsymbol{a}_1, \boldsymbol{r}_1, \ldots, \boldsymbol{s}_T, \boldsymbol{a}_T, \boldsymbol{r}_T \rangle$. Slightly abuse of notations, we also use $\boldsymbol{\omega} \in \mathcal{D}$ to represent $\boldsymbol{\omega}$ is in some trajectories of $\mathcal{D}$. To simplify the conditional generation process, we construct the target trajectory fragment $\boldsymbol{x}(\tau)$, which is a consecutive sub-sequence of trajectory $\tau$, along with the essential conditional information $\boldsymbol{y}(\tau)$ as

$$\boldsymbol{x}(\tau) = \begin{bmatrix} \boldsymbol{s}_t & \boldsymbol{s}_{t+1} & \cdots & \boldsymbol{s}_{t+H-1} \\ \boldsymbol{a}_t & \boldsymbol{a}_{t+1} & \cdots & \boldsymbol{a}_{t+H-1} \end{bmatrix}, \qquad \boldsymbol{y}(\tau) = [\boldsymbol{\omega}, \boldsymbol{\omega} \odot \boldsymbol{R}(\tau)], \tag{3}$$

where $\odot$ denotes the element-wise product, $\boldsymbol{\omega} \odot \boldsymbol{R}(\tau) = \sum_t \gamma^t \boldsymbol{\omega} \odot \mathcal{R}(\boldsymbol{s}_t, \boldsymbol{a}_t)$ is the weighted vector-valued return, and $H$ is the predefined horizon. For notation simplicity, we use $\boldsymbol{x}, \boldsymbol{y}, \boldsymbol{R}$ to denote $\boldsymbol{x}(\tau), \boldsymbol{y}(\tau), \boldsymbol{R}(\tau)$. By optimizing Equation 2, we obtain a conditional distribution estimator $p_\theta$ to generate trajectory fragments $\boldsymbol{x}$ according to the given preference $\boldsymbol{\omega}$ and maximize the vector-valued return $\boldsymbol{\omega}^\top \boldsymbol{R} = \boldsymbol{1}^\top(\boldsymbol{\omega} \odot \boldsymbol{R})$. Specifically, trajectory fragments are generated through the reverse denoising process of the diffusion model:

$$p_\theta(\boldsymbol{x}_0|\boldsymbol{y}) = \int p(\boldsymbol{x}_K) \prod_{k=1}^{K} p_\theta(\boldsymbol{x}_{k-1}|\boldsymbol{x}_k, \boldsymbol{y}) \mathrm{d}\boldsymbol{x}_{1:K}, \tag{4}$$

which is implemented as an iterative denoising process via a noise prediction network $\epsilon_\theta(\boldsymbol{x}_k, \boldsymbol{y}, k)$ trained by minimizing the simplified objective in Equation 1.

## 4.2 TRAINING WITH MIXUP-SYNTHESIZED TRAJECTORIES

Diffusion models are highly expressive and can accurately generate in-distribution trajectories after training on the original dataset. To ensure these models learn the underlying trajectory distribution rather than simply memorizing the trajectories, DIFFMORL employs the mixup (Zhang et al., 2018) technique to mitigate the memorization phenomenon and facilitate feature learning with a modified optimization objective, thereby improving the generalization on OOD preferences.

**Mixup-based Augmented Learning Process**    DIFFMORL applies the mixup technique to linearly interpolate the original trajectories and synthesize additional pseudo-trajectories. Specifically, before updating the diffusion model, a training batch $\{(\boldsymbol{\omega}_i, \boldsymbol{x}_i, \boldsymbol{R}_i)\}_{i=1}^b$ is randomly drawn from the dataset $\mathcal{D}$, where $b$ is the batch size. Then, two sub-batches are drawn from this batch as $\{(\boldsymbol{\omega}_j^1, \boldsymbol{x}_j^1, \boldsymbol{R}_j^1)\}_{j=1}^{b'}$ and $\{(\boldsymbol{\omega}_j^2, \boldsymbol{x}_j^2, \boldsymbol{R}_j^2)\}_{j=1}^{b'}$. A random coefficient $\lambda \sim U(-\lambda_0, 1 + \lambda_0)$, where $\lambda_0 > 0$, is used to linearly combine the two sub-batches to produce new samples:

$$
\begin{aligned}
\tilde{\boldsymbol{\omega}}_j &= \lambda \boldsymbol{\omega}_j^1 + (1 - \lambda) \boldsymbol{\omega}_j^2 \\
\tilde{\boldsymbol{x}}_j &= \lambda \boldsymbol{x}_j^1 + (1 - \lambda) \boldsymbol{x}_j^2 \qquad \text{for } j = 1, \ldots, b' \\
\tilde{\boldsymbol{R}}_j &= \lambda \boldsymbol{R}_j^1 + (1 - \lambda) \boldsymbol{R}_j^2
\end{aligned}
\tag{5}
$$

These new samples are inserted into the original batch for training the diffusion model:

$$
\text{New batch} = \{(\boldsymbol{\omega}_i, \boldsymbol{x}_i, \boldsymbol{R}_i)\}_{i=1}^b \cup \{(\tilde{\boldsymbol{\omega}}_j, \tilde{\boldsymbol{x}}_j, \tilde{\boldsymbol{R}}_j)\}_{j=1}^{b'}.
\tag{6}
$$

Note that we allow the coefficient $\lambda$ to be negative or exceed 1 to enable extrapolation. Additionally, to prevent the excessive influence of the pseudo-trajectories, employing appropriate early stopping for mixup-based training at the $N'$-th step of the total $N$ training steps is advantageous. A detailed study of of the corresponding hyperparameters is provided in Appendix A.2.

**Overall Training Objective**    The DIFFMORL framework is trained in a self-supervised manner, where samples are drawn from the dataset, augmented with mixup, and diffused with Gaussian noises, i.e., the forward process. The goal is to predict the noises based on target information, i.e., the reverse denoising process. We modify the original loss function in Equation 1 for training as follows:

$$
\mathcal{L}(\theta) = \mathbb{E}_{\epsilon, k, \tau \sim mixup(\mathcal{D}), \beta \sim \text{Bern}(p)} \left[ \|\epsilon - \epsilon_\theta(\boldsymbol{x}_k; \boldsymbol{\omega}, (1 - \beta)\boldsymbol{\omega} \odot \boldsymbol{R} + \beta \varnothing, k)\|^2 \right],
\tag{7}
$$

where $\epsilon \sim \mathcal{N}(\boldsymbol{0}, \boldsymbol{I})$ is the target noise, $k$ is the diffusion timestep uniformly sampled from $\{1, \ldots, K\}$, $\tau \sim mixup(\mathcal{D})$ represents trajectories sampled from the dataset $\mathcal{D}$ and then augmented with mixup as Equation 6, and $\beta \sim \text{Bern}(p)$ is a Bernoulli random variable used for blocking the condition $\boldsymbol{\omega} \odot \boldsymbol{R}$ with probability $p$. We parameterize the noise prediction network as a conditional U-Net (Ronneberger et al., 2015), with extended modules for conditioning. The architecture design and more details are provided in Appendix A. After training on the pre-collected dataset with Equation 7, DIFFMORL is capable of accurately generating desired trajectories corresponding to diverse in-distribution and OOD preferences, which are utilized for planning and online task execution in the next section.

## 4.3 PLANNING AND EXECUTION WITH CONDITIONAL GENERATION

Here, we introduce how DIFFMORL realizes planning and online execution given a preference during deployment. Specifically, DIFFMORL must control the trajectory generation process to produce a plan $\boldsymbol{x}$ that aligns with the preference $\boldsymbol{\omega}$, maximizes the scalarized return $\boldsymbol{\omega}^\top \boldsymbol{R}$, and remains consistent with the real states. We design the following techniques to achieve these goals.

**Independent Preference Encoding**    Unlike previous works (Zhu et al., 2023a) on offline MORL that make decisions on $\boldsymbol{x}' = [\boldsymbol{x}, \boldsymbol{\omega}]$ by concatenating trajectory fragments with preferences and encoding them with a single encoder, DIFFMORL processes them separately, utilizing an independent MLP encoder to encode preferences. The reason is that these two elements possess very different modalities. Trajectory fragments are more varied and of high frequency, even within a single trajectory, while preferences remain stationary throughout each episode. By using separate encoders, DIFFMORL can more effectively capture the distinct features of each element, leading to a better matching between the generated trajectories and the given preferences.

**Weighted Vector-valued Return Guidance**    To further improve the quality of the generated trajectory fragments, DIFFMORL must properly set the return-vector conditions to guide the generation process. To this end, we calculate the maximal value ever achieved by the behavior policies for each objective from the dataset, denoted as $R_i^{\max}$, which serves as an estimation of the maximum value for the $i$-th objective. We then construct a pseudo-return $\boldsymbol{R}^{\max} = [R_1^{\max}, \ldots, R_n^{\max}]$ to guide the generation process of DIFFMORL. To emphasize the varying importance of different objectives according to a given preference $\boldsymbol{\omega}$, we re-weight the pseudo-return with the preference as $\boldsymbol{\omega} \odot \boldsymbol{R}^{\max}$. Finally, classifier-free diffusion guidance is applied with the following noise estimation:

$$\hat{\epsilon} = \epsilon_\theta(\boldsymbol{x}_k; \boldsymbol{\omega}, \varnothing, k) + w\left[\epsilon_\theta(\boldsymbol{x}_k; \boldsymbol{\omega}, \boldsymbol{\omega} \odot \boldsymbol{R}^{\max}, k) - \epsilon_\theta(\boldsymbol{x}_k; \boldsymbol{\omega}, \varnothing, k)\right], \tag{8}$$

where $w$ is the guidance scale to balance the diversity and quality of the generated trajectory fragments.

**Consistent Planning and Execution**    After setting the condition mechanism based on the given preference and return vector, DiffMORL can generate a trajectory fragment through the iterative denoising process from $\boldsymbol{x}_K \sim \mathcal{N}(\boldsymbol{0}, \boldsymbol{I})$ for $K$ steps. To ensure the generated trajectory fragment begins at the agent's current state $s_t$, i.e., consistent planning, DIFFMORL replaces the first noisy state in $\boldsymbol{x}_k(k = 1, \ldots, K)$ with the ground-truth state $\boldsymbol{s}_t$, then denoises the remaining portion of the trajectory fragment. Upon finishing the denoising process, DIFFMORL extracts the first generated action $\boldsymbol{a}_t$ for online execution, transitioning the environment to the next state, receiving a vector-valued reward, and advancing the MORL task.

With the well-designed model architecture, training objective, and conditioning mechanism, DIFFMORL can effectively learn from the offline dataset and complete MORL tasks in an online manner.

## 5 EXPERIMENTS

In this section, we conduct extensive experiments on D4MORL (Zhu et al., 2023a) to answer the following questions: (1) How will DIFFMORL benefit generalization? (Section 5.2) (2) Can DIFFMORL outperforms baselines on both complete and incomplete datasets? (Section 5.3) (3) Can DIFFMORL generalize well on different levels of incompleteness? (Section 5.4) (4) How different components affect the performance of DIFFMORL? (Section 5.5)

### 5.1 D4MORL BENCHMARK AND METRICS

**Setup and Baselines**    In our experiment, we consider offline MORL tasks of the Datasets for Multi-Objective Reinforcement Learning (D4MORL) benchmark (Zhu et al., 2023a). D4MORL is based on six multi-objective MuJoCo (Todorov et al., 2012) environments, including five environments with two objectives each (MO-Ant, MO-HalfCheetah, MO-Hopper, MO-Swimmer, MO-Walker2d) and one with three objectives (MO-Hopper-3obj). It features a variety of datasets that differ in tasks, data quality (`Expert` or `Amateur`), and preference ranges (`High-H`, `Med-H`, or `Low-H`). To better evaluate generalization, we additionally collect `incomplete` datasets containing preference-lacking regions as illustrated in Section 4.1 by reject sampling using behavior policies. These regions can be described by *centers* and *radii*. After training on these datasets, all methods are tested on 324 (MO-Hopper-3obj) or 500 (other environments) equally spaced preference points in $\Omega$.

We include various categories of offline MORL algorithms as baselines, including imitation learning by behavior cloning BC(P), conservative offline RL method CQL(P) (Kumar et al., 2020), sequential modeling methods MODT(P) and MORvS(P)(Zhu et al., 2023a[1]) and diffusion based method MODULI (Yuan et al., 2024). Note that all of the baselines except MODULI, concatenate preferences with trajectory fragments as $\boldsymbol{x}'(\tau) = [\boldsymbol{x}(\tau), \boldsymbol{\omega}]$ for the MORL setting. For more details of the environments, datasets and baselines, please refer to Appendix B.

**Metrics**    To evaluate the performances of different multi-objective algorithms on competing objectives, we must introduce the notion of *Pareto Optimality*. We refer to the solution $\boldsymbol{G}^{\pi_p}$ to be *dominated* by $\boldsymbol{G}^{\pi_q}$, denoted as $\boldsymbol{G}^{\pi_p} \prec \boldsymbol{G}^{\pi_q}$, if $\boldsymbol{G}_i^{\pi_p} \leq \boldsymbol{G}_i^{\pi_q}, \forall i \in \{1, \ldots, n\}$ and $\boldsymbol{G}^{\pi_p} \neq \boldsymbol{G}^{\pi_q}$. All optimal (in the sense of dominance) solutions form the *Pareto Front*, denoted as $P$. In MORL,

---

[1] In this work, we focus on the preference-conditioned version of the baselines, which performs better than the non-conditioned version, and omit the "(P)" symbols in the following for notation simplicity.

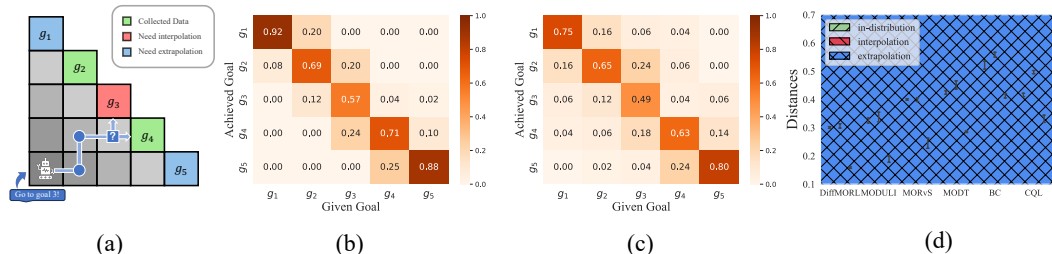

(a)        (b)        (c)        (d)

Figure 2: A case study in a grid navigation task. (a) The overview of the navigation task. (b) The probability distribution heatmap of achieved goals versus given goals of DIFFMORL (c) The probability distribution heatmap of achieved goals versus given goals of MORvS (d) The distances between in-distribution and OOD outputs distributions and optimal probability distributions.

the goal is to derive a policy such that its empirical Pareto front is a good approximation of the Pareto front. Since the true Pareto front for many problems is unknown, two metrics (Hayes et al., 2022) for relative comparisons on empirical Pareto front $P$ among different algorithms will be used: **Hypervolume (Hv)** $:= \int_{\mathbb{R}^n} \mathbf{1}_{H(P)}(\boldsymbol{z}) \mathrm{d}\boldsymbol{z}$, where $H(P) = \{\boldsymbol{z} \in \mathbb{R}^n \mid \exists \boldsymbol{p} \in P, \boldsymbol{p}_0 \prec \boldsymbol{z} \prec \boldsymbol{p}\}$, $\boldsymbol{p}_0$ is a predefined reference point, and $\mathbf{1}_{H(P)}(\boldsymbol{z})$ is the indicator function. Larger Hv means larger volume of space that is enclosed by the Pareto front and coordinate planes, and the better. **Sparsity (Sp)** $:= \frac{1}{|P|-1} \sum_{i=1}^{n} \sum_{k=1}^{|P|-1} [\tilde{P}_i(k) - \tilde{P}_i(k+1)]^2$, where $\tilde{P}_i(k)$ is the $k$-th value in the sorted list for the $i$-th objective values of $P$. Smaller Sp means denser approximation of the Pareto front, and the better when given close Hv. To evaluate the generalization ability of different algorithms on OOD preferences, we design a new metric termed **Return Mismatch (Rm)** $:= \sum_{\boldsymbol{p} \in P} \|\boldsymbol{G}^*(\boldsymbol{\omega}(\boldsymbol{p})) - \boldsymbol{p}\|_1$, where $\boldsymbol{\omega}(\boldsymbol{p})$ is the preference of the solution $\boldsymbol{p}$, $\boldsymbol{G}^*(\boldsymbol{\omega})$ is the optimal solution for preference $\boldsymbol{\omega}$, approximated by one expert solution $\boldsymbol{R}(\hat{\boldsymbol{\omega}})$ with the closest preference approximation $\hat{\boldsymbol{\omega}}$ and maximal vectorized return $\hat{\boldsymbol{\omega}}^\top \boldsymbol{R}(\hat{\boldsymbol{\omega}})$. Smaller Rm represents better approximation of the Pareto front at the preference-lacking regions, i.e., better generalization. We run each method for three distinct seeds to calculate the mean $\pm$ standard error of the metrics.

## 5.2 CASE STUDY

To gain deeper insight into how diffusion models facilitate generalization, we conduct experiments on a simple yet illustrative task shown in Figure 2(a). In this task, an agent is located at the lower left corner of a grid world, and is requested to navigate to one of the five goals $g_1, \ldots, g_5$ by moving upward($U$) or rightward($R$). We first train the agent with trajectories end at $g_2$ and $g_4$ generated with random policy, and then we request it to reach $g_3$ (which needs interpolation generalization) or $g_1, g_5$ (which need extrapolation generalization). The results of the achieved goals versus given goals tested on DIFFMORL and MORvS are shown in Figure 2(b) and 2(c) in the form of probability distribution matrcies as well as heatmaps, revealing that DIFFMORL with deeper main diagonal achieves better in-distribution performance and OOD generalization compared with MORvS with shallower color. To further assess the ability of different methods' performance and different types of generalization, we calculated three metrics based on the results of the matrices of achieved goals versus given goals by defining distances for in-distribution performance: $\frac{1}{2} \sum_{i \in \{2,4\}} D_{TV}(G_{:,i} \| I_{:,i})$, interpolation generalization: $D_{TV}(G_{:,3} \| I_{:,3})$ and extrapolation generalization: $\frac{1}{2} \sum_{i \in \{1,5\}} D_{TV}(G_{:,i} \| I_{:,i})$, where $G$ is the probability matrices, $I$ is the identity matrix that stands for the optimal matrix and $D_{TV}$ is the total variance distance. Note that the sum of the two generalization distance metrics is analogous to the **Return Mismatch** metric we introduced in Section 5.1, both measuring the generalization gap. The results are listed in Figure 2(d), where DIFFMORL achieves the best in-distribution performance and OOD generalization among others.

Essentially, we argue that diffusion process and mixup facilitate generalization by mixing and learning the distributions of trajectory fragments. For example, agents may reach $g_2$ by acting $RU + UU$. Through the learning process of DIFFMORL, trajectory fragments $RU$ and $UU$ are effectively extracted by applying mixup, learned and composed by the diffusion model. Agent thus can perform $RU + RU$ to reach $g_3$, or perform $UU + UU$ to reach $g_1$, achieving both types of generalization.

Table 1: Mean $\pm$ standard error of Hv and Sp on `High-H-Expert` datasets. $\uparrow$ means the higher is the better, and $\downarrow$ means the lower is the better. Entries with zero sparsity are omitted. (Dataset: performance of the behavioral policies estimated based on the dataset. "Best Count" in the tables means the times one algorithm outperforms the others in terms of mean metric value.)

| Environments | Metrics | Dataset | DIFFMORL | MODULI | MORvS | MODT | BC | CQL |
|---|---|---|---|---|---|---|---|---|
| MO-Ant | Hv $(\times 10^6)$ ↑ | 6.39 | 6.37 ± 0.03 | **6.39 ± 0.02** | 6.37 ± 0.03 | 6.07 ± 0.33 | 4.85 ± 0.34 | 5.98 ± 0.13 |
| | Sp $(\times 10^4)$ ↓ | \ | **0.71 ± 0.31** | 0.79 ± 0.12 | 0.81 ± 0.29 | 1.80 ± 0.89 | 5.06 ± 2.12 | 4.32 ± 1.92 |
| MO-HalfCheetah | Hv $(\times 10^6)$ ↑ | 5.79 | **5.79 ± 0.00** | **5.79 ± 0.00** | 5.78 ± 0.00 | 5.74 ± 0.03 | 5.65 ± 0.02 | 5.64 ± 0.05 |
| | Sp $(\times 10^4)$ ↓ | \ | **0.06 ± 0.01** | 0.07 ± 0.00 | 0.07 ± 0.03 | 0.10 ± 0.02 | 0.16 ± 0.06 | 0.20 ± 0.13 |
| MO-Hopper | Hv $(\times 10^7)$ ↑ | 2.09 | 2.07 ± 0.01 | **2.09 ± 0.01** | 1.98 ± 0.05 | 1.96 ± 0.03 | 1.50 ± 0.18 | 1.66 ± 0.01 |
| | Sp $(\times 10^5)$ ↓ | \ | **0.08 ± 0.02** | 0.09 ± 0.01 | 0.35 ± 0.17 | 0.31 ± 0.07 | 6.39 ± 5.08 | 4.17 ± 0.34 |
| MO-Hopper-3obj | Hv $(\times 10^{10})$ ↑ | 3.82 | **3.62 ± 0.10** | 3.57 ± 0.02 | 3.39 ± 0.13 | 3.05 ± 0.23 | 2.18 ± 0.37 | 0.75 ± 0.21 |
| | Sp $(\times 10^5)$ ↓ | \ | 0.19 ± 0.05 | **0.07 ± 0.00** | 0.32 ± 0.03 | 0.26 ± 0.01 | 0.39 ± 0.41 | 0.19 ± 0.10 |
| MO-Swimmer | Hv $(\times 10^4)$ ↑ | 3.26 | **3.25 ± 0.00** | 3.24 ± 0.00 | 3.22 ± 0.00 | 3.24 ± 0.00 | 3.19 ± 0.01 | 3.20 ± 0.10 |
| | Sp $(\times 10^0)$ ↓ | \ | 4.17 ± 1.27 | 4.43 ± 0.38 | 6.76 ± 2.14 | 6.43 ± 3.98 | 13.36 ± 8.69 | **1.28 ± 0.26** |
| MO-Walker2d | Hv $(\times 10^6)$ ↑ | 5.22 | **5.20 ± 0.00** | **5.20 ± 0.00** | 5.10 ± 0.03 | 5.10 ± 0.02 | 3.57 ± 0.30 | 2.92 ± 0.41 |
| | Sp $(\times 10^4)$ ↓ | \ | **0.10 ± 0.01** | 0.11 ± 0.01 | 0.46 ± 0.14 | 0.43 ± 0.10 | 18.93 ± 16.19 | 1.42 ± 0.23 |
| Best Count (total=12) | | \ | **8** | 5 | 0 | 0 | 0 | 1 |

Table 2: Mean $\pm$ standard error of Hv, Sp and Rm on `incomplete High-H-Expert` datasets.

| Environments | Metrics | Dataset | DIFFMORL | MODULI | MORvS | MODT | BC | CQL |
|---|---|---|---|---|---|---|---|---|
| MO-Ant | Hv $(\times 10^6)$ ↑ | 6.26 | 6.32 ± 0.06 | 6.38 ± 0.02 | **6.41 ± 0.01** | 6.13 ± 0.11 | 4.87 ± 0.61 | 5.79 ± 0.38 |
| | Sp $(\times 10^4)$ ↓ | \ | **0.79 ± 0.13** | 0.86 ± 0.08 | 1.08 ± 0.42 | 1.03 ± 0.52 | 3.29 ± 2.92 | 3.68 ± 0.28 |
| | Rm $(\times 10^2)$ ↓ | \ | **2.10 ± 0.14** | 2.20 ± 0.20 | 2.27 ± 0.50 | 5.62 ± 3.42 | 5.83 ± 0.50 | 8.73 ± 0.37 |
| MO-HalfCheetah | Hv $(\times 10^6)$ ↑ | 5.63 | **5.69 ± 0.00** | 5.68 ± 0.01 | 5.64 ± 0.01 | 5.61 ± 0.02 | 5.51 ± 0.03 | 5.46 ± 0.21 |
| | Sp $(\times 10^4)$ ↓ | \ | **0.16 ± 0.06** | 0.18 ± 0.07 | 0.29 ± 0.03 | 0.39 ± 0.04 | 1.30 ± 0.39 | 0.24 ± 0.04 |
| | Rm $(\times 10^2)$ ↓ | \ | **1.92 ± 0.31** | 2.32 ± 0.20 | 3.27 ± 0.11 | 3.28 ± 0.08 | 5.01 ± 0.04 | 6.12 ± 0.17 |
| MO-Hopper | Hv $(\times 10^7)$ ↑ | 2.07 | **2.05 ± 0.01** | 2.01 ± 0.00 | 2.00 ± 0.03 | 1.77 ± 0.06 | 0.97 ± 0.57 | 1.37 ± 0.18 |
| | Sp $(\times 10^5)$ ↓ | \ | 0.39 ± 0.08 | **0.18 ± 0.02** | 0.90 ± 0.38 | 2.08 ± 2.42 | 5.37 ± 5.85 | 1.87 ± 0.25 |
| | Rm $(\times 10^3)$ ↓ | \ | **2.46 ± 0.80** | 2.52 ± 0.36 | 2.73 ± 0.31 | 3.88 ± 0.04 | 5.87 ± 2.65 | 3.67 ± 0.91 |
| MO-Hopper-3obj | Hv $(\times 10^{10})$ ↑ | 3.73 | **3.46 ± 0.18** | 3.40 ± 0.15 | 2.97 ± 0.36 | 2.47 ± 0.17 | 2.31 ± 0.25 | 0.72 ± 0.18 |
| | Sp $(\times 10^5)$ ↓ | \ | 0.17 ± 0.01 | **0.13 ± 0.01** | 0.22 ± 0.11 | 0.26 ± 0.02 | 0.24 ± 0.04 | 0.30 ± 0.09 |
| | Rm $(\times 10^3)$ ↓ | \ | 2.99 ± 0.12 | 2.46 ± 0.19 | 1.93 ± 0.28 | 2.86 ± 0.13 | **1.26 ± 0.40** | 3.73 ± 0.84 |
| MO-Swimmer | Hv $(\times 10^4)$ ↑ | 3.21 | 3.24 ± 0.01 | 3.24 ± 0.01 | 3.22 ± 0.00 | **3.24 ± 0.00** | 2.99 ± 0.33 | 3.02 ± 0.03 |
| | Sp $(\times 10^0)$ ↓ | \ | 5.68 ± 0.70 | 5.76 ± 0.45 | 6.51 ± 3.21 | 5.09 ± 1.11 | 110 ± 157 | **1.59 ± 0.17** |
| | Rm $(\times 10^0)$ ↓ | \ | **5.92 ± 2.28** | 6.01 ± 1.74 | 11.21 ± 4.27 | 39.46 ± 19.44 | 48.56 ± 54.26 | 56.06 ± 4.38 |
| MO-Walker2d | Hv $(\times 10^6)$ ↑ | 5.07 | **5.12 ± 0.02** | 5.10 ± 0.00 | 5.05 ± 0.01 | 4.99 ± 0.03 | 3.69 ± 0.05 | 2.90 ± 0.34 |
| | Sp $(\times 10^4)$ ↓ | \ | **0.21 ± 0.04** | 0.29 ± 0.02 | 0.47 ± 0.08 | 0.63 ± 0.25 | 8.67 ± 2.17 | 1.17 ± 0.31 |
| | Rm $(\times 10^2)$ ↓ | \ | **7.62 ± 1.17** | 8.33 ± 1.36 | 13.45 ± 3.35 | 15.19 ± 5.32 | 26.07 ± 0.83 | 25.93 ± 5.28 |
| Best Count (total=18) | | \ | **12** | 2 | 1 | 1 | 1 | 1 |

## 5.3 COMPETITIVE RESULTS

We first compare DIFFMORL with baseline methods on the `High-H-Expert` datasets, which have complete and uniform preference coverage, in all six environments. The results are shown in Table 1. We observe that the widely used CQL method and the simple method BC produce sub-optimal policies on most tasks due to their over-conservatism and less expressive MLP backbone when facing multi-objective tasks. On the other hand, both sequential modeling methods MORvS and MODT exhibit similar performances, achieving near-optimal results in most environments. Similar to our method, MODULI applys expressive diffusion models and explicitly handles OOD preferences, which performs relatively well. Whilst our approach, DIFFMORL, performs comparably well or exceeds MODULI, and also outperforms other baselines due to its more accurate generation, which is demonstrated by its lower Sp on most tasks. Furthermore, DIFFMORL achieves Hv very close to the behavioral policies with relatively low variance, indicating its effectiveness and stability on learning offline MORL datasets with complete preference coverage.

To evaluate the generalization ability of different algorithms, we extend the above experiment with the Rm metric to *incomplete* datasets. In Table 2, we find that although these baselines perform well on a few tasks, they still struggle for performance due to over-conservatism, limited expressiveness or relatively inaccurate preference understanding. However, DIFFMORL enhances its generalization and generation accuracy by the mixup training and conditioned generation respectively, and performs

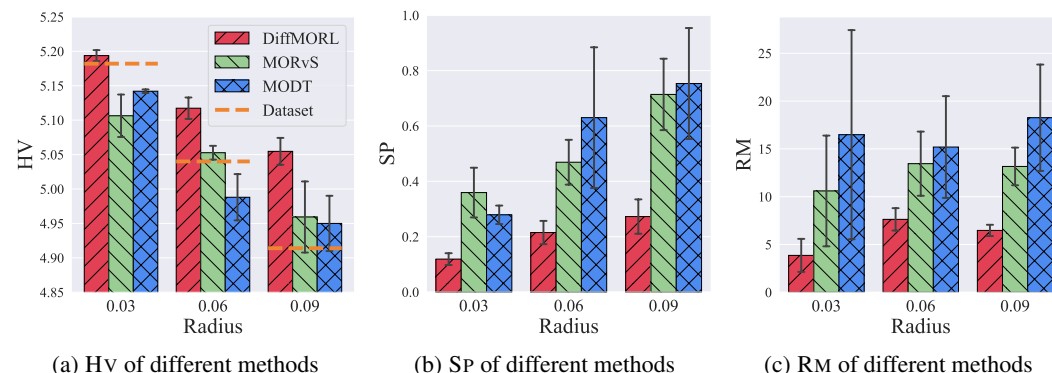

(a) Hᴠ of different methods    (b) Sᴘ of different methods    (c) Rᴍ of different methods

Figure 4: Performance on different levels of `incomplete High-H-Expert` datasets of the MO-Walker2d environment. Scales: Hᴠ $\times 10^6$, Sᴘ $\times 10^3$, Rᴍ $\times 10^2$.

the best among baselines. Remarkably, Dɪғғ MORL surpasses other baseline on 8 of the 12 metrics on complete datasets, and on 12 of the 18 metrics on incomplete datasets, underscoring its remarkable generalization ability. The full results are deferred to Appendix D.3.

As an illustrative example, we visualize the Pareto fronts of the `High-H-Expert` and `incomplete High-H-Expert` datasets of MO-HalfCheetah, alongside the empirical Pareto fronts of Dɪғғ MORL and the best baseline MORᴠS in Figure 3. Note that the positions of the four Pareto fronts almost overlap, and we slightly shift them for visual clarity. Also, we allow a small tolerance for displaying the dominated solutions. Compared to the dataset (◎) with even coverage, the incomplete dataset (●) lacks trajectories in the upper right region of the Pareto front, which corresponds to the OOD preferences. When learning from the incomplete dataset, both methods perform well for in-distribution preferences (● and ●). However, MORᴠS fails to generalize, as evidenced by its inability to cover the preference-lacking region (●). In contrast, Dɪғғ MORL successfully produces correct and near-optimal trajectories for the OOD preferences (●), effectively completing the preference-lacking region. More visualization are given in Appendix D.4.

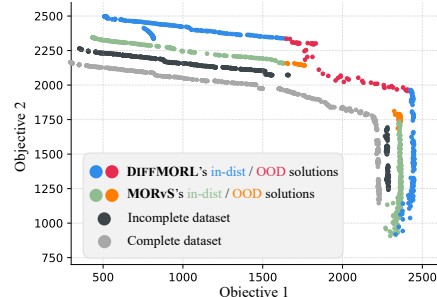

Figure 3: An example of the Pareto fronts.

### 5.4 Generalization and Performance on Different Levels of Incompleteness

To investigate Dɪғғ MORL's performance on various levels of incompleteness, we control the sizes, i.e., the *radii*, of the preference-lacking regions in `incomplete High-H-Expert` datasets of the MO-Walker2d environment. This approach generates several new generalization tasks, with increasing incompleteness corresponding to larger radii. As shown in Figure 4, the task becomes more challenging as the dataset becomes more incomplete, indicated by the performance decrease of all methods with increasing radius. Notably, Dɪғғ MORL consistently outperforms MORᴠS and MODT across all three metrics. Furthermore, as the radius increases, the advantages of Dɪғғ MORL over other methods gradually increases. This demonstrates Dɪғғ MORL's robust performance across different levels of dataset incompleteness. Additionally, we examine the impact of varying the positions of the preference-lacking regions and list the numerical results in Appendix D.2.

### 5.5 Ablation Study

The two main components designed for promoting the generalization of Dɪғғ MORL are mixup-based training (MT, in contrast to conventional training without mixup, CT) and independent preference encoding (IPE, in contrast to preference concatenation with trajectory fragments, PC). In this section, we conduct an ablation study on the `incomplete High-H-Expert` dataset of the MO-HalfCheetah environment to study how these two components affect the generalization ability of Dɪғғ MORL

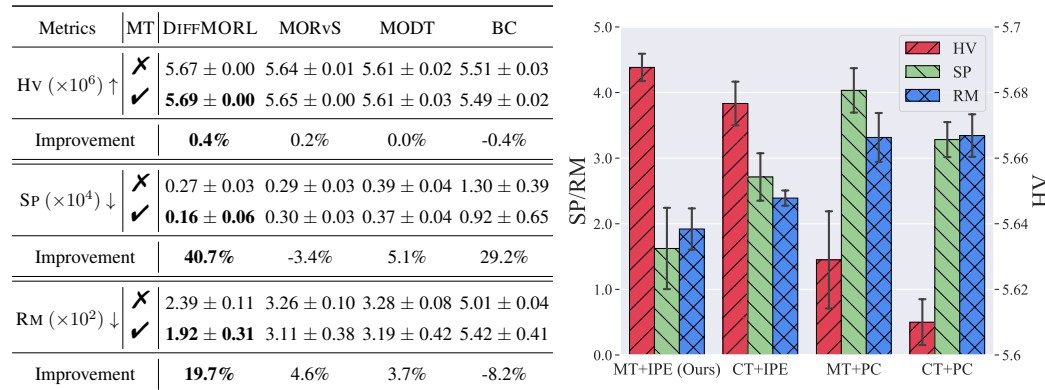

Figure 5: **Left**: Mean ± standard error of Hv, Sp and Rm on `incomplete High-H-Expert` datasets of the MO-HalfCheetah environment. **Right**: Performance of DiffMORL equipped with different components. MT: Mixup-based Training, CT: Conventional Training, IPE: Independent Preference Encoding, PC: Preference Concatenation. Scales: Hv $\times 10^6$, Sp $\times 10^3$, Rm $\times 10^2$.

and other baselines. As listed in the left table of Figure 5, regardless of whether MT is utilized, DiffMORL consistently achieves the best performance. Furthermore, when equipped with MT, DiffMORL demonstrates the most significant performance improvement among all methods. In contrast, other baselines show very limited performance improvement from MT, such as MORvS and MODT, or even suffer performance degradation, as seen with BC. We hypothesize that this is due to the relatively lower expressiveness and generalization ability of the backbones in these methods. This validates that the mixup technique needs to be paired with models with strong expressiveness, like diffusion models, to maximize its effectiveness.

To analyse the joint effect of MT and IPE on promoting the generalization of DiffMORL, we control their use in the training and evaluation pipeline, obtaining results shown in the right part of Figure 5. We find that without either of these techniques, DiffMORL suffers from performance degradation. Additionally, the Rm metric indicates that DiffMORL equipped with IPE benefits more from MT in terms of generalization. On the other hand, without the accurate preference understanding provided by IPE, MT leads to higher variance and degradation in performance and generalization, as evidenced by the Hv and Sp metrics.

We further show the necessity of applying mixup data augmentation for extracing trajectory fragments and preventing memorization instead of other simpler data augmentation like injecting noise to the trajectories. Recall that in Equation 6 we augment incomplete datasets by synthesize new trajectories with mixup. Here, we instead add or multiply trajectory data with truncated Gaussian noise to produce new trajectories. The results is shown in Table 7 in Appendix D.1, revealing that mixup is necessary for the generalization of DiffMORL, while other data augmentation methods provide limited promotion in generalization. In summary, we conclude that mixup-based training and independent preference encoding, essentially work holistically for promoting the generalization of DiffMORL.

## 6 Final Remarks

In this work, we propose DiffMORL, a diffusion-based framework, equipped with mixup-based training and independent preference encoding, for generalizable offline MORL. Leveraging the strong generation and generalization capability of diffusion models, DiffMORL can generate near-optimal plans and generalize well on out-of-distribution preferences. We conduct extensive experiments on the D4MORL benchmark and intuitively demonstrate the performance and generalization capabilities of DiffMORL. Further ablation study reveals that diffusion-based model, mixup-based training and independent preference encoding are the keys for generalizable planning in offline MORL tasks. In future research, we will delve into deeper aspects of generalization properties of diffusion models, and further improve generalization on broader tasks such as multi-agent reinforcement learning. We further discuss the limitations and potential improvements of DiffMORL in Appendix C.

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

# A DETAILS OF DIFFMORL

## A.1 ARCHITECTURE

We implement DIFFMORL based on the widely adopted diffuser framework (Janner et al., 2022; Ajay et al., 2023), where the noise prediction network is parameterized with U-Net (Ronneberger et al., 2015), and several MLPs are used for encoding conditions. As depicted in Figure 6, before the entire procedure begins, an agent interacts with the environment and obtains multi-objective data labeled with preferences and returns. DIFFMORL first loads the data and augments it using mixup to extend the data range. The augmented trajectory fragments are then noised and fed into the diffusion model along with the corresponding preferences and returns. The diffusion model predicts the noises added to the samples.

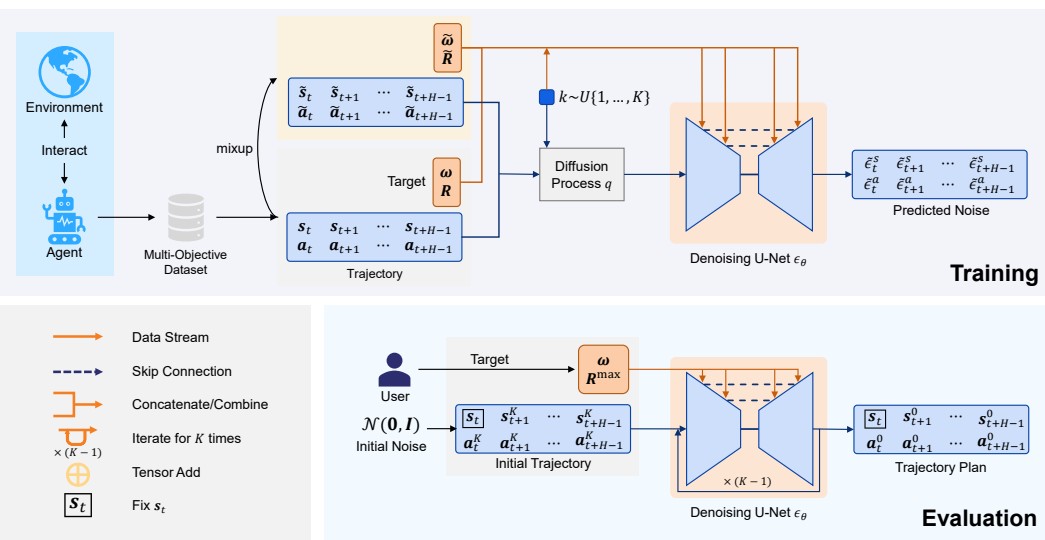

Figure 6: The architecture of DIFFMORL

After training, DIFFMORL can be leveraged for multi-objective planning, where a user specifies a target preference while aiming to maximize the scalarized return. At first, a trajectory fragment is initialized as Gaussian noise, with the first state fixed to the ground truth state. This trajectory fragment and the target are fed into the diffusion model for $K$ iterations of denoising.

Once the denoising process is done, the diffusion model produces a trajectory plan, and the first action is extracted for execution. Following reward and state transitioning may arrive, and DIFFMORL continues to generate trajectory plan based on new current state and extract the next action to execute. Besides, we modified the structure of the residual temporal block in the U-Net, as shown in Figure 7. Specifically, we utilize two additional MLP encoders to encode the preference and vector-valued return conditions. The embeddings of diffusion timestep and both conditions are concatenated

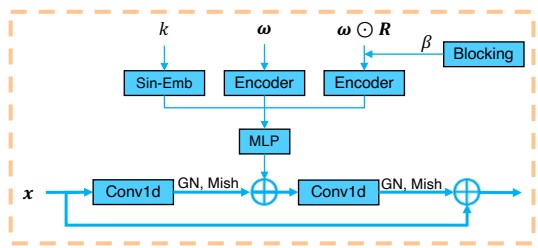

Figure 7: Residual temporal block in the U-Net

and fed into an MLP, and then added to the embeddings of trajectory fragments. "Blocking" is for blocking the condition with some probability to train the classifier-free diffusion guidance. In evaluation, the "Blocking" operation is disabled.

Our code implementation is based on PEDA(Zhu et al., 2023a) (`https://github.com/baitingzbt/PEDA/`) and Decision-Diffuser(Ajay et al., 2023) (`https://github.com/anuragajay/decision-diffuser/`).

## A.2 HYPERPARAMETERS

Table 3: Generic hyperparameters of DIFFMORL.

| Hyperparameter | Value |
|---|---|
| Condition Encoder | FC(64, 256, 64) with Mish activations |
| Learning Rate | $2 \times 10^{-4}$ |
| Weight Decay | $1 \times 10^{-4}$ |
| Optimizer | AdamW |
| Batch Size $b$ | 32 |
| Diffusion Step $K$ | 8 |
| Maximum Trajectory Length $T$ | 500 |
| Horizon $H$ | 8 |
| $\lambda_0$ | 0.5 |
| $p$ (Bernoulli parameter in Equation. 7) | 0.1 |

Table 4: Hyperparameters of DIFFMORL for different datasets.

| Environment | Quality | Guidance Scale $w$ | mixup Number $b'$ | mixup Step $N'(\times 10^4)$ | Training Step $N(\times 10^4)$ |
|---|---|---|---|---|---|
| MO-Ant | Expert | 0.1 | 8 | 10 | 10 |
| | Amateur | 0.1 | 6 | 10 | 10 |
| MO-HalfCheetah | Expert | 0.1 | 6 | 40 | 40 |
| | Amateur | 1 | 6 | 20 | 20 |
| MO-Hopper | Expert | 0.1 | 6 | 5 | 40 |
| | Amateur | 0.1 | 6 | 20 | 30 |
| MO-Hopper-3obj | Expert | 0.1 | 5 | 10 | 20 |
| | Amateur | 0.1 | 5 | 10 | 10 |
| MO-Swimmer | Expert | 0.1 | 5 | 10 | 20 |
| | Amateur | 0.1 | 5 | 5 | 5 |
| MO-Walker2d | Expert | 0.1 | 6 | 15 | 40 |
| | Amateur | 1 | 6 | 10 | 10 |

We use the generic hyperparameters shown in Table 3 for all experiments, and we finetune the guidance scale $w$, mixup number $b'$, mixup early stopping step $N'$ and total training step $N$ on every environment in D4MORL benchmark, and choose that with the highest hypervolume, as shown in Figure 4. Note that it is still possible to apply more careful finetuning on the guidance scale and total training step, to obtain even higher performance and generalization on Amateur quality datasets. Furthermore, we analyse the sensitivity to the hyperparameters of mixup-base training: $b'$, $N'$ and

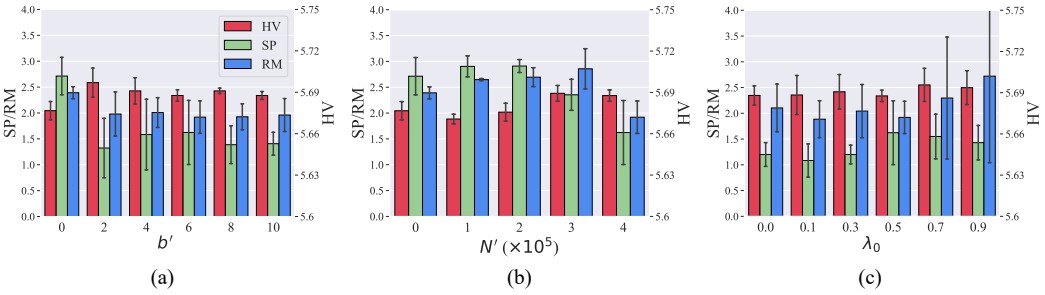

Figure 8: Sensitivity to (a) mixup number $b'$, (b) mixup early stopping step $N'$ and (c) parameter $\lambda_0$ in mixup-based training. The error bars are the standard errors across 3 different seeds. Scales: HV $\times 10^6$, SP $\times 10^3$, RM $\times 10^2$.

$\lambda_0$, as shown in Figure 8. The experiments are carried out on the incomplete High-H dataset of the MO-HalfCheetah environment. The results show that DIFFMORL is stable to $b'$, $N'$, as the

standard errors are small, and there is no significant deviation in means. While for $\lambda_0$, a smaller value $(0.1 \sim 0.5)$ is preferred as it leads to more stable and higher performance and generalization. We additionally test the case where $\lambda_0 = 0$, which performs slightly worse than $0.1 \sim 0.5$, due to the limited extrapolation. To summarize Figure 8, we argue that DIFFMORL is **stable** to these three hyperparameters, indicating a stable performance and generalization ability of DIFFMORL.

### A.3 PSEUDO CODES

In this section, we outline the training and planning procedure of DIFFMORL in Algorithm 1 and Algorithm 2. In the training pipeline, our goal is to train the noise prediction network of the diffusion model using the dataset. We first sample a batch of data from the dataset, and augment it through the mixup technique. Then, we sample a noise, a random diffusion step and a blocking variable to train the noise prediction network by minimizing the loss function in Equation 7 till converge.

After training, we can utilize DIFFMORL for planning: First, the agent observe current state $s_t$, and DIFFMORL samples the initial noisy trajectory fragment. Then DIFFMORL starts the denoising process and denoise the noisy trajectory fragment for $K$ steps, using the state information $s_t$, target information $y$ and classifier-free guidance (Ho & Salimans, 2021). Upon finishing the denoising process, the action $a_t$ is extracted from the generated trajectory plan $x_0$ and executed, producing reward $r_t$ and transitioning the environment to next state $s_{t+1}$. This procedure continues until the decision making process is done.

---

**Algorithm 1:** Train DIFFMORL

---

**Input:** Dataset $\mathcal{D}$, diffusion timestep $K$, horizon $H$, history length $h$, $\lambda_0$, Bernoulli parameter $p$
**Result:** Noise predictor $\epsilon_\theta$
Initialize $\epsilon_\theta$ and its optimizer
**while** *not converge* **do**
    Get a batch of trajectories $\tau$ with horizon $H$ from $\mathcal{D}$
    // Augment the dataset with mixup
    Sample $\lambda \sim U(-\lambda_0, 1 + \lambda_0)$
    Produce new synthetic samples $\tilde{\tau}$ as Equation 5 and combine: $\tau' = \tau \cup \tilde{\tau}$
    // Train the diffusion model
    Sample noise $\epsilon \sim \mathcal{N}(\mathbf{0}, \boldsymbol{I})$, diffusion timestep $k \sim U(\{1, \ldots, K\})$, $\beta \sim \text{Bern}(p)$
    Optimize $\epsilon_\theta$ by minimizing $\mathcal{L}(\theta)$ in Equation 7, with $\epsilon, k, \tau', \beta$
**end**

---

**Algorithm 2:** Plan with DIFFMORL

---

**Input:** Noise predictor $\epsilon_\theta$, diffusion timestep $K$, horizon $H$, guidance scale $w$, condition $y$,
        precomputed $\boldsymbol{R}^{\max}$
Initialize time step $t = 0$, set the generation length of $\epsilon_\theta$ to $H$
**while** *not done* **do**
    Observe current state $s_t$, initialize $\boldsymbol{x}_K \sim \mathcal{N}(\mathbf{0}, \boldsymbol{I})$
    // Denoise for $K$ steps
    **for** $k = K, \ldots, 1$ **do**
        // Construct necessary conditions
        Replace the first state of $\boldsymbol{x}_k$ to be consistent with current state $s_t$
        Construct $\boldsymbol{\omega}, \boldsymbol{\omega} \odot \boldsymbol{R}^{\max}$ from $y$
        // Classifier-free guidance
        Obtain $\hat{\epsilon} = \epsilon_\theta(\boldsymbol{x}_k; \boldsymbol{\omega}, \varnothing, k) + w\left[\epsilon_\theta(\boldsymbol{x}_k; \boldsymbol{\omega}, \boldsymbol{\omega} \odot \boldsymbol{R}^{\max}, k) - \epsilon_\theta(\boldsymbol{x}_k; \boldsymbol{\omega}, \varnothing, k)\right]$
        Denoise $\boldsymbol{x}_k$ with $\hat{\epsilon}$ and obtain $\boldsymbol{x}_{k-1}$
    **end**
    // Extract the first action for execution
    Extract $a_t$ from $\boldsymbol{x}_0$
    Execute $a_t$, obtain reward $r_t$ and transition to $s_{t+1}$
    $t \leftarrow t + 1$
**end**

---

## A.4 COMPUTE RESOURCES

We run our experiments on GeForce RTX 2080 Ti. A typical training of $4 \times 10^5$ steps takes about 12 hours, and planning with 8 diffusion timesteps, for 500 different trajectories each with maximal length of 500 takes about 10 hours. Besides, there should be at least 32GB memory and 32GB storage space to run any single experiment successfully. At least 360GB storage space is needed for maintaining all datasets at the same time.

## B DETAILS OF ENVIRONMENTS, DATA COLLECTION AND BASELINES

### B.1 ENVIRONMENTAL SETTINGS

Here we list some important information of each environment, including the main objectives that are specialized in each environment, and the state and action dimension in Table 5. For more details and implementations of these environment, please refer to the literatures (Zhu et al., 2023a; Xu et al., 2020).

Table 5: Main information of D4MORL environments.

| Environment | Objectives | Dimensions |
|---|---|---|
| MO-Ant | x-axis speed, y-axis speed | $\mathcal{S} \subseteq \mathbb{R}^{27}, \mathcal{A} \subseteq \mathbb{R}^8$ |
| MO-HalfCheetah | forward speed, energy efficiency | $\mathcal{S} \subseteq \mathbb{R}^{17}, \mathcal{A} \subseteq \mathbb{R}^6$ |
| MO-Hopper | forward speed, jumping height | $\mathcal{S} \subseteq \mathbb{R}^{11}, \mathcal{A} \subseteq \mathbb{R}^3$ |
| MO-Hopper-3obj | forward speed, jumping height, energy efficiency | $\mathcal{S} \subseteq \mathbb{R}^{11}, \mathcal{A} \subseteq \mathbb{R}^3$ |
| MO-Swimmer | forward speed, energy efficiency | $\mathcal{S} \subseteq \mathbb{R}^8 , \mathcal{A} \subseteq \mathbb{R}^2$ |
| MO-Walker2d | forward speed, energy efficiency | $\mathcal{S} \subseteq \mathbb{R}^{17}, \mathcal{A} \subseteq \mathbb{R}^6$ |

### B.2 INCOMPLETE DATA COLLECTION

Datasets in D4MORL benchmark vary in environment, data quality and preference range. However, D4MORL considers only the width of the preference coverage, which implies a contiguous Pareto front, and that is why we call this kind of preference coverage as "preference range". We argue that preference range provided in D4MORL are either too wide (`High-H`, `Med-H`) or too narrow (`Low-H`) so that the generalization of different methods cannot differentiate from each other upon evaluations.

In our setting, we further consider preference coverage that implies a Pareto front with gaps. We implement a new module that enables creating gaps by reject sampling based on the preference range that D4MORL provides, and thus add a new attribute `incomplete` to each dataset in D4MORL, *allowing for more nuanced comparison in generalization ability*. For example, rejecting all samples with $\boldsymbol{\omega} \in \{\boldsymbol{\omega}' \mid \|\boldsymbol{\omega}' - [0.5, 0.5]\|_1 \leq 0.1 \times 2)\}$ based on `High-H` datasets produces `incomplete` `High-H` datasets that are lacking in demonstrations of preferences around $[0.5, 0.5]$, or specifically, preferences between $[0.4, 0.6]$ and $[0.6, 0.4]$ are lacking. In this example, the *center* is $\boldsymbol{\omega} = [0.5, 0.5]$ and the *radius* is $0.1$. Note that the approach for reject sampling here is consistent with the formulation in Section 4.1, hence the incomplete datasets are exactly the cases we focus on. Considering of the space, time and the problem of the width of preference range in D4MORL, we only collect incomplete datasets for `High-H` ones and evaluate on them. Details of incomplete datasets for each environment in our experiments are shown in Table 6. Preference-lacking regions with more centers and distinct radii are also supported in our code.

Table 6: The parameters of preference-lacking regions of `incomplete High-H-Expert` datasets used in our experiments.

| Environment | Center | Radius |
|---|---|---|
| MO-Ant | $[0.5, 0.5]$ | 0.06 |
| MO-HalfCheetah | $[0.5, 0.5]$ | 0.06 |
| MO-Hopper | $[0.45, 0.55]$ | 0.04 |
| MO-Hopper-3obj | $[1/3, 1/3, 1/3]$ | 0.04 |
| MO-Swimmer | $[0.5, 0.5]$ | 0.06 |
| MO-Walker2d | $[0.5, 0.5]$ | 0.06 |

### B.3 DETAILS OF BASELINES

In this section, we describe the details of the baselines:

- **MODT** is a direct extension of the widely used Decision Transformer (DT) (Chen et al., 2021a), which encodes states $s_t$, actions $a_t$ and return-to-go (RTG) $g_t = \sum_{t'=t}^{T} r_{t'}$ as tokens. These tokens represents a trajectory $\tau = \langle s_1, a_1, g_1, \ldots, s_T, a_T, g_T \rangle$ that can be processed by causally masked transformer architecture such as GPT (Radford et al., 2019). MODT additionally concatenate preference vectors with states, actions and RTG as $s^* = [s, \omega], a^* = [a, \omega], g^* = [g, \omega]$ and form new trajectory $\tau^*$ for decision making. Besides, MODT also inputs the preference-weighted RTG $g_t \odot \omega$ for stable training.

- **MORvS** can be seen as a variant of MODT, which conditions on carefully selected conditions to further promote its performance (Emmons et al., 2022). In contrast to MODT, MORvS concatenate the preference with the states and the average RTGs, and encode everything as one single input.

- **MODULI** is a diffusion-based planning framework similar to our method which also applies diffusion models for generalizable MORL. Different from our work, MODULI proposes a sliding guidance mechanism to facilitate generalization, where a plug-and-play slider adapter is trained to encode preference variation. It also parameterizes the backbones of diffusion models with DiT (Peebles & Xie, 2023) instead of Unet(Ronneberger et al., 2015).

- **BC(P)** simply uses supervised loss to train the policy network that directly maps the states (concatenated with preferences) to actions. The policy network of BC(P) is parameterized with MLP and runs very fast compared to MODT. Note that BC(P) do not use reward information.

- **CQL(P)** is the multi-objective version of the state-of-the-art single objective offline RL method Conservative Q-Learning (Kumar et al., 2020), which learns a conservative Q-function $f : \mathcal{S} \times \mathcal{A} \to \mathbb{R}$ to lower-bounds the true value and is suitable for tasks with complex and multi-modal data distributions. Based on CQL, CQL(P) modifies the network architecture and takes preference vectors as inputs to learn a preference-conditioned Q-function $f^* : \mathcal{S} \times \mathcal{A} \times \Omega \to \mathbb{R}$.

We train these baselines for $4 \times 10^5$ steps each. We use the MODT, MORvS and multi-objective version BC implemented in `https://github.com/baitingzbt/PEDA/`, and we implemented multi-objective CQL according to the instructions in D4MORL literature (Zhu et al., 2023a) based on the CQL implementations in `https://github.com/zhyang2226/DMBP/`. We follow the instructions in Yuan et al. (2024) to implement MODULI. The policies of BC and CQL are parameterized with MLPs. All hyperparameters are consistent with the default settings in D4MORL.

## C DISCUSSIONS

### C.1 LIMITATIONS

Diffusion models are mainly hindered by their slow sampling originated from their iterative denoising process, which limits the application of DIFFMORL for control and planning tasks that require high-frequency response in real world. For instance, despite our best efforts to reduce sampling time, the decision process of DIFFMORL in MO-HalfCheetah environment takes about 0.18s wall-clock

Table 7: Mean $\pm$ standard error of Hv, Sp and Rm of different data augmentation methods on `incomplete High-H-Expert` datasets of the MO-HalfCheetah environment.

|  | MT (Ours) | Add | Multiply | No augmentation |
|---|---|---|---|---|
| Hv ($\times 10^6$) $\uparrow$ | **5.69$\pm$0.00** | $5.66 \pm 0.01$ | $5.67 \pm 0.01$ | $5.67 \pm 0.00$ |
| Sp ($\times 10^4$) $\downarrow$ | **0.16$\pm$0.06** | $0.25 \pm 0.06$ | $0.24 \pm 0.03$ | $0.23 \pm 0.03$ |
| Rm ($\times 10^2$) $\downarrow$ | **1.92$\pm$0.31** | $2.36 \pm 0.15$ | $2.36 \pm 0.07$ | $2.39 \pm 0.11$ |

time to generate one trajectory plan and extract the first action to execute. To further accelerate sampling without loss of performance, more advanced models such as consistency models (Song et al., 2023; Chen et al., 2024) could be utilized.

### C.2 POTENTIAL IMPROVEMENTS

There is possibility that DIFFMORL can be applied to a broader range of utility functions, as we do not put much assumption on the form of it. Specifically, for the linear utility function $f(\boldsymbol{\omega}, \boldsymbol{r}) = \boldsymbol{\omega}^\top \boldsymbol{r}$ we considered, it can be expressed in a more informative vector form $\boldsymbol{\omega} \odot \boldsymbol{r}$ rather than the less informative scalar form $\boldsymbol{\omega}^\top \boldsymbol{r} = \mathbf{1}^\top (\boldsymbol{\omega} \odot \boldsymbol{r})$. We argue that the more informative "weighted vector-valued return" further enhances the ability of DIFFMORL to accurately understand preferences and expected returns, ultimately leading to near-optimal trajectory plans. This insight may be helpful for other multi-objective tasks with different forms of utility functions.

## D EXTENSIVE RESULTS

### D.1 RESULTS OF DIFFERENT DATA AUGMENTATION METHODS

We conduct an experiment on `incomplete High-H-Expert` datasets of MO-HalfCheetah environments by replacing the data augmentation methods of DIFFMORL with additive or multiplicative noise, instead of the original mixup. In practice, we generate new trajectories by add or multiply the real trajectories from the dataset with truncated Gaussian noise of mean 0 (for adding) or 1 (for multiplying) and variance 0.01, truncated to $[-0.1, 0.1]$. The results in Table 7 shows that mixup in DIFFMORL is necessary for the generalization and cannot be replaced by noise injection.

### D.2 RESULTS ON DIFFERENT LEVELS OF INCOMPLETENESS

To further investigate the generalization of different methods on different levels of incompleteness, we control the *Center* ans *Radius* of the `incomplete High-H-Expert` dataset of the MO-Walker2d environment to produce several tasks, and sort the tasks from the hardest to the easiest according to the corresponding Hv of the datasets. According to Table 8, DIFFMORL consistently outperforms all baselines in all tasks and all metrics. For MODULI, despite its near optimal Hv, it is inferior compared with DIFFMORL in terms of Rm, due to the lack of mixup training. Importantly, the Hv's of DIFFMORL are even higher than that of the datasets, while baselines can hardly or never do. From the results of the Sp and Rm metrics, we can see that DIFFMORL significantly outperforms baselines, indicating the best ability among baselines to approximate the Pareto front and to generalize to OOD preferences. To summarize Table 8 we conclude that DIFFMORL exhibits remarkable performance and generalization ability, both agnostic to the incompleteness level.

### D.3 RESULTS ON D4MORL DATASETS

This section presents the full results of DIFFMORL and all baselines evaluated on all D4MORL datasets and the extended incomplete datasets, containing different environments, data quality and preference coverage. The results are shown in Table 9, Table 10 and Table 11 for hypervolume, sparsity and return mismatch metrics respectively. All results are reported as mean $\pm$ standard error across three different seeds. "Best Count" in the tables means the times one algorithm outperforms the others in terms of mean metric value. Here `incomplete` stands for `incomplete High-H` dataset of each environment. Since sometimes more than one methods achieves the same best performance,

Table 8: Mean ± standard error of Hv, Sp and Rm on different levels of incompleteness.

| Center | Radius | Metrics | Dataset | DIFFMORL | MODULI | MORvS | MODT | BC | MOCQL |
|---|---|---|---|---|---|---|---|---|---|
| [0.5, 0.5] | 0.09 | Hv ($\times10^6$) ↑ | 4.914 | **5.06 ± 0.02** | 5.01 ± 0.01 | 4.96 ± 0.05 | 4.95 ± 0.04 | 3.38 ± 0.42 | 2.73 ± 0.02 |
| | | Sp ($\times10^4$) ↓ | \ | **0.27 ± 0.06** | 0.32 ± 0.05 | 0.71 ± 0.13 | 0.75 ± 0.20 | 14.33 ± 21.22 | 1.07 ± 0.58 |
| | | Rm ($\times10^2$) ↓ | \ | **6.47 ± 0.59** | 9.17 ± 0.68 | 13.16 ± 1.97 | 18.26 ± 5.56 | 21.37 ± 1.65 | 26.32 ± 1.38 |
| [0.5, 0.5] | 0.06 | Hv ($\times10^6$) ↑ | 5.04 | **5.12 ± 0.02** | 5.10 ± 0.00 | 5.05 ± 0.01 | 4.99 ± 0.03 | 3.69 ± 0.05 | 2.90 ± 0.34 |
| | | Sp ($\times10^4$) ↓ | \ | **0.21 ± 0.04** | 0.29 ± 0.02 | 0.47 ± 0.08 | 0.63 ± 0.25 | 8.68 ± 2.18 | 1.17 ± 0.31 |
| | | Rm ($\times10^2$) ↓ | \ | **7.62 ± 1.17** | 8.33 ± 1.36 | 13.45 ± 3.36 | 15.19 ± 5.32 | 26.07 ± 0.83 | 26.93 ± 5.28 |
| [0.6, 0.4] | 0.06 | Hv ($\times10^6$) ↑ | 5.14 | **5.18 ± 0.01** | 5.15 ± 0.02 | 5.01 ± 0.09 | 5.14 ± 0.03 | 2.48 ± 0.86 | 3.05 ± 0.32 |
| | | Sp ($\times10^4$) ↓ | \ | 0.13 ± 0.01 | **0.10 ± 0.01** | 0.78 ± 0.38 | 0.31 ± 0.15 | 23.25 ± 24.96 | 1.56 ± 0.20 |
| | | Rm ($\times10^2$) ↓ | \ | **2.98 ± 0.72** | 3.56 ± 0.51 | 12.29 ± 4.35 | 8.88 ± 1.41 | 22.32 ± 1.86 | 21.37 ± 4.13 |
| [0.4, 0.6] | 0.06 | Hv ($\times10^6$) ↑ | 5.149 | **5.17 ± 0.00** | 5.13 ± 0.01 | 5.10 ± 0.03 | 5.09 ± 0.03 | 3.16 ± 0.49 | 3.34 ± 0.10 |
| | | Sp ($\times10^4$) ↓ | \ | 0.27 ± 0.03 | 0.32 ± 0.02 | 0.37 ± 0.17 | **0.27 ± 0.01** | 3.12 ± 2.88 | 0.45 ± 0.20 |
| | | Rm ($\times10^2$) ↓ | \ | 8.35 ± 1.27 | **6.44 ± 0.68** | 13.77 ± 4.23 | 14.05 ± 4.61 | 17.98 ± 4.30 | 15.62 ± 2.23 |
| [0.5, 0.5] | 0.03 | Hv ($\times10^6$) ↑ | 5.182 | **5.19 ± 0.01** | **5.19 ± 0.01** | 5.11 ± 0.03 | 5.14 ± 0.00 | 2.86 ± 0.27 | 3.67 ± 0.87 |
| | | Sp ($\times10^4$) ↓ | \ | **0.12 ± 0.02** | 0.14 ± 0.02 | 0.36 ± 0.09 | 0.28 ± 0.03 | 2.32 ± 2.73 | 0.50 ± 0.23 |
| | | Rm ($\times10^2$) ↓ | \ | **3.86 ± 1.73** | 4.28 ± 1.20 | 10.60 ± 5.79 | 16.48 ± 10.94 | 27.11 ± 2.88 | 20.31 ± 3.42 |
| Best Count (total=15) | | | \ | **12** | 3 | 0 | 1 | 0 | 0 |

the sum of Best Count across all methods may exceed the number of metrics on different tasks. The same conclusion can be obtained from the full results, that DIFFMORL outperforms all baselines significantly, in terms of performance and generalization ability.

Table 9: The full results on Hypervolume metric

| Environments | Quality | Range | Behavior | DIFFMORL | MODULI | MORvS | MODT | BC | CQL |
|---|---|---|---|---|---|---|---|---|---|
| MO-Ant ($\times10^6$) | Expert | High-H | 6.39 | 6.37 ± 0.03 | **6.39 ± 0.02** | 6.37 ± 0.03 | 6.08 ± 0.34 | 4.85 ± 0.34 | 5.98 ± 0.13 |
| | | Med-H | 6.44 | **6.40 ± 0.01** | 6.38 ± 0.01 | 6.35 ± 0.02 | 6.22 ± 0.03 | 5.10 ± 0.26 | 6.05 ± 0.16 |
| | | Low-H | 5.26 | 5.61 ± 0.17 | 5.55 ± 0.10 | 5.17 ± 0.06 | 5.42 ± 0.08 | 5.07 ± 0.10 | **6.01 ± 0.10** |
| | | incomplete | 6.26 | 6.32 ± 0.06 | 6.38 ± 0.02 | **6.41 ± 0.01** | 6.13 ± 0.11 | 4.87 ± 0.61 | 5.79 ± 0.38 |
| | Amateur | High-H | 5.60 | 5.98 ± 0.16 | 6.08 ± 0.03 | **6.10 ± 0.04** | 0.03 ± 0.01 | 4.44 ± 0.26 | 5.68 ± 0.21 |
| | | Med-H | 5.67 | 5.94 ± 0.10 | 5.90 ± 0.06 | **6.04 ± 0.05** | 3.19 ± 2.99 | 4.27 ± 0.30 | 5.72 ± 0.24 |
| | | Low-H | 5.26 | 5.15 ± 0.18 | 5.10 ± 0.06 | 5.04 ± 0.05 | 0.12 ± 0.08 | 4.65 ± 0.08 | **5.60 ± 0.11** |
| | | incomplete | 5.59 | 5.81 ± 0.18 | 5.76 ± 0.17 | **6.06 ± 0.02** | 0.37 ± 0.30 | 4.31 ± 0.29 | 5.62 ± 0.24 |
| MO-HalfCheetah ($\times10^6$) | Expert | High-H | 5.79 | **5.79 ± 0.00** | **5.79 ± 0.00** | 5.78 ± 0.00 | 5.74 ± 0.03 | 5.66 ± 0.02 | 5.64 ± 0.05 |
| | | Med-H | 5.79 | **5.79 ± 0.00** | **5.79 ± 0.00** | 5.76 ± 0.01 | 5.60 ± 0.16 | 5.60 ± 0.11 | 5.65 ± 0.03 |
| | | Low-H | 4.75 | **4.92 ± 0.03** | 4.87 ± 0.04 | 4.91 ± 0.03 | 4.83 ± 0.05 | 4.75 ± 0.03 | 4.89 ± 0.08 |
| | | incomplete | 5.63 | **5.69 ± 0.00** | 5.68 ± 0.01 | 5.64 ± 0.01 | 5.60 ± 0.02 | 5.51 ± 0.03 | 5.46 ± 0.21 |
| | Amateur | High-H | 5.70 | 5.74 ± 0.01 | 5.76 ± 0.00 | **5.78 ± 0.00** | 5.59 ± 0.01 | 5.66 ± 0.02 | 5.56 ± 0.03 |
| | | Med-H | 5.69 | 5.71 ± 0.03 | 5.77 ± 0.00 | **5.79 ± 0.01** | 5.59 ± 0.01 | 5.48 ± 0.04 | 5.57 ± 0.04 |
| | | Low-H | 4.14 | 4.67 ± 0.11 | 4.70 ± 0.02 | 4.76 ± 0.01 | 4.44 ± 0.34 | **4.78 ± 0.04** | 4.72 ± 0.06 |
| | | incomplete | 5.42 | **5.65 ± 0.02** | 5.64 ± 0.01 | 5.63 ± 0.01 | 5.60 ± 0.01 | 5.49 ± 0.11 | 5.48 ± 0.13 |
| MO-Hopper ($\times10^7$) | Expert | High-H | 2.09 | 2.07 ± 0.01 | **2.09 ± 0.01** | 1.96 ± 0.05 | 1.98 ± 0.05 | 1.50 ± 0.18 | 1.66 ± 0.01 |
| | | Med-H | 2.09 | 2.04 ± 0.03 | **2.05 ± 0.01** | 1.92 ± 0.07 | 1.92 ± 0.02 | 1.04 ± 0.90 | 1.25 ± 0.12 |
| | | Low-H | 1.80 | **1.76 ± 0.00** | 1.73 ± 0.01 | 1.72 ± 0.03 | 1.69 ± 0.07 | 0.80 ± 0.70 | 0.98 ± 0.36 |
| | | incomplete | 2.07 | **2.05 ± 0.01** | 2.01 ± 0.00 | 2.00 ± 0.03 | 1.77 ± 0.06 | 0.97 ± 0.57 | 1.37 ± 0.18 |
| | Amateur | High-H | 2.01 | 1.95 ± 0.06 | **2.01 ± 0.01** | 1.80 ± 0.08 | 1.64 ± 0.07 | 1.37 ± 0.36 | 1.73 ± 0.03 |
| | | Med-H | 1.98 | **1.94 ± 0.05** | 1.90 ± 0.02 | 1.79 ± 0.01 | 1.59 ± 0.19 | 0.97 ± 0.85 | 1.60 ± 0.05 |
| | | Low-H | 1.73 | **1.76 ± 0.04** | 1.73 ± 0.01 | 1.58 ± 0.08 | 1.50 ± 0.08 | 0.53 ± 0.56 | 1.02 ± 0.34 |
| | | incomplete | 1.99 | **1.92 ± 0.10** | 1.86 ± 0.03 | 1.79 ± 0.02 | 1.58 ± 0.04 | 1.25 ± 0.22 | 1.37 ± 0.24 |
| MO-Hopper-3obj ($\times10^{10}$) | Expert | High-H | 3.82 | **3.62 ± 0.10** | 3.57 ± 0.02 | 3.39 ± 0.13 | 3.05 ± 0.23 | 2.18 ± 0.37 | 0.75 ± 0.21 |
| | | Med-H | 3.71 | 3.43 ± 0.07 | **3.48 ± 0.03** | 3.23 ± 0.17 | 2.87 ± 0.15 | 1.94 ± 0.17 | 0.66 ± 0.18 |
| | | Low-H | 0.95 | 0.96 ± 0.05 | 1.03 ± 0.05 | **1.20 ± 0.19** | 1.15 ± 0.18 | 0.00 ± 0.00 | 0.60 ± 0.12 |
| | | incomplete | 3.73 | **3.46 ± 0.18** | 3.40 ± 0.15 | 2.97 ± 0.36 | 2.47 ± 0.17 | 2.31 ± 0.25 | 0.72 ± 0.18 |
| | Amateur | High-H | 3.34 | 2.79 ± 0.27 | **3.33 ± 0.06** | 2.69 ± 0.18 | 1.38 ± 0.12 | 1.84 ± 0.31 | 0.66 ± 0.42 |
| | | Med-H | 3.06 | 2.12 ± 0.15 | 2.48 ± 0.08 | **2.51 ± 0.23** | 1.04 ± 0.09 | 1.41 ± 0.85 | 0.71 ± 0.31 |
| | | Low-H | 1.01 | 0.88 ± 0.38 | 1.06 ± 0.32 | **1.31 ± 0.22** | 0.63 ± 0.22 | 1.26 ± 0.20 | 0.56 ± 0.32 |
| | | incomplete | 3.23 | 2.47 ± 0.19 | 2.51 ± 0.10 | **2.53 ± 0.03** | 1.28 ± 0.23 | 1.88 ± 0.07 | 0.68 ± 0.38 |
| MO-Swimmer ($\times10^4$) | Expert | High-H | 3.26 | **3.25 ± 0.00** | 3.24 ± 0.00 | 3.22 ± 0.00 | 3.24 ± 0.00 | 3.19 ± 0.01 | 3.20 ± 0.10 |
| | | Med-H | 3.26 | **3.24 ± 0.00** | **3.24 ± 0.00** | 3.22 ± 0.01 | 3.24 ± 0.01 | 3.14 ± 0.12 | 3.18 ± 0.08 |
| | | Low-H | 2.47 | 2.70 ± 0.02 | 2.56 ± 0.03 | **2.83 ± 0.10** | 2.53 ± 0.02 | 2.66 ± 0.06 | 2.73 ± 0.02 |
| | | incomplete | 3.21 | 3.24 ± 0.01 | 3.24 ± 0.01 | 3.22 ± 0.00 | **3.24 ± 0.00** | 2.99 ± 0.33 | 3.02 ± 0.03 |
| | Amateur | High-H | 2.13 | 3.17 ± 0.01 | **3.20 ± 0.00** | 2.77 ± 0.05 | 0.64 ± 0.05 | 2.76 ± 0.04 | 1.76 ± 0.34 |
| | | Med-H | 2.14 | 3.16 ± 0.03 | **3.18 ± 0.01** | 2.73 ± 0.05 | 0.65 ± 0.05 | 2.76 ± 0.04 | 1.74 ± 0.15 |
| | | Low-H | 1.69 | **2.85 ± 0.09** | 2.76 ± 0.05 | 2.52 ± 0.10 | 0.63 ± 0.03 | 2.37 ± 0.06 | 1.21 ± 0.13 |
| | | incomplete | 2.17 | **3.17 ± 0.02** | 2.68 ± 0.16 | 2.30 ± 0.38 | 0.62 ± 0.03 | 2.75 ± 0.04 | 1.68 ± 0.32 |
| MO-Walker2d ($\times10^6$) | Expert | High-H | 5.22 | **5.20 ± 0.00** | **5.20 ± 0.00** | 5.10 ± 0.03 | 5.10 ± 0.02 | 3.57 ± 0.30 | 2.92 ± 0.41 |
| | | Med-H | 5.22 | **5.20 ± 0.00** | 5.19 ± 0.00 | 5.11 ± 0.01 | 4.99 ± 0.05 | 2.71 ± 0.56 | 2.86 ± 0.26 |
| | | Low-H | 4.55 | **4.56 ± 0.04** | 4.56 ± 0.06 | 4.54 ± 0.03 | 3.78 ± 0.14 | 0.94 ± 1.63 | 2.65 ± 0.39 |
| | | incomplete | 5.07 | **5.12 ± 0.00** | 5.10 ± 0.00 | 5.05 ± 0.01 | 4.99 ± 0.03 | 3.69 ± 0.05 | 2.90 ± 0.34 |
| | Amateur | High-H | 5.02 | 4.93 ± 0.16 | **5.06 ± 0.00** | 5.06 ± 0.01 | 2.97 ± 0.35 | 3.96 ± 0.15 | 3.68 ± 0.37 |
| | | Med-H | 5.03 | 5.01 ± 0.06 | **5.03 ± 0.03** | 5.02 ± 0.04 | 2.94 ± 1.00 | 3.86 ± 0.06 | 3.72 ± 0.76 |
| | | Low-H | 4.47 | 4.45 ± 0.03 | 4.44 ± 0.02 | **4.46 ± 0.12** | 2.84 ± 1.61 | 3.59 ± 0.16 | 3.64 ± 0.68 |
| | | incomplete | 4.87 | **5.07 ± 0.00** | 4.98 ± 0.02 | 4.88 ± 0.01 | 3.08 ± 0.25 | 3.55 ± 0.44 | 3.32 ± 0.45 |
| Best Count (total=48) | | | \ | **22** | 14 | 12 | 1 | 1 | 2 |

Table 10: The full results on Sparsity metric. Zero Sparsity entries are omitted.

| Environments | Quality | Range | DIFFMORL | MODULI | MORvS | MODT | BC | CQL |
|---|---|---|---|---|---|---|---|---|
| MO-Ant ($\times 10^4$) | Expert | High-H | **0.71 ± 0.31** | 0.79 ± 0.12 | 0.81 ± 0.29 | 1.80 ± 0.89 | 5.06 ± 2.12 | 4.32 ± 1.92 |
| | | Med-H | 0.76 ± 0.13 | 0.74 ± 0.10 | **0.73 ± 0.10** | 0.94 ± 0.32 | 3.41 ± 1.55 | 4.06 ± 1.39 |
| | | Low-H | 1.05 ± 0.31 | 0.85 ± 0.20 | 0.76 ± 0.12 | **0.60 ± 0.19** | 1.29 ± 1.35 | 2.18 ± 0.29 |
| | | incomplete | **0.79 ± 0.13** | 0.86 ± 0.08 | 1.08 ± 0.42 | 1.03 ± 0.52 | 3.29 ± 2.92 | 3.68 ± 0.28 |
| | Amateur | High-H | 1.10 ± 0.39 | **0.53 ± 0.05** | 0.85 ± 0.11 | 0.00 ± 0.00 | 1.91 ± 1.71 | 4.98 ± 2.10 |
| | | Med-H | 1.07 ± 0.26 | 0.83 ± 0.12 | 0.72 ± 0.10 | **0.43 ± 0.38** | 3.90 ± 5.70 | 4.22 ± 1.69 |
| | | Low-H | 1.17 ± 0.79 | 1.21 ± 0.32 | 0.90 ± 0.64 | 0.00 ± 0.00 | **0.49 ± 0.14** | 1.56 ± 0.38 |
| | | incomplete | 1.18 ± 0.69 | 1.33 ± 0.46 | **0.98 ± 0.20** | 5.40 ± 4.88 | 1.32 ± 0.56 | 4.92 ± 0.21 |
| MO-HalfCheetah ($\times 10^4$) | Expert | High-H | **0.06 ± 0.01** | 0.07 ± 0.00 | 0.07 ± 0.03 | 0.10 ± 0.02 | 0.15 ± 0.05 | 0.20 ± 0.13 |
| | | Med-H | **0.06 ± 0.02** | 0.07 ± 0.00 | 0.07 ± 0.01 | 0.09 ± 0.05 | 0.18 ± 0.12 | 0.24 ± 0.07 |
| | | Low-H | 0.15 ± 0.07 | 0.19 ± 0.03 | 0.21 ± 0.04 | 0.08 ± 0.05 | **0.05 ± 0.01** | 0.06 ± 0.01 |
| | | incomplete | **0.16 ± 0.06** | 0.18 ± 0.07 | 0.29 ± 0.03 | 0.39 ± 0.05 | 1.31 ± 0.40 | 0.24 ± 0.04 |
| | Amateur | High-H | 0.12 ± 0.03 | **0.07 ± 0.02** | 0.14 ± 0.18 | 0.08 ± 0.01 | 0.09 ± 0.05 | 0.12 ± 0.05 |
| | | Med-H | 0.23 ± 0.27 | 0.14 ± 0.03 | **0.05 ± 0.01** | 0.10 ± 0.01 | 0.26 ± 0.05 | 0.23 ± 0.06 |
| | | Low-H | 0.07 ± 0.05 | 0.04 ± 0.00 | 0.04 ± 0.05 | 0.03 ± 0.02 | **0.02 ± 0.02** | 0.03 ± 0.02 |
| | | incomplete | 0.24 ± 0.11 | 0.21 ± 0.03 | 0.21 ± 0.04 | **0.09 ± 0.00** | 0.34 ± 0.07 | 0.22 ± 0.06 |
| MO-Hopper ($\times 10^5$) | Expert | High-H | **0.08 ± 0.02** | 0.09 ± 0.01 | 0.35 ± 0.17 | 0.31 ± 0.07 | 6.39 ± 5.08 | 4.17 ± 0.34 |
| | | Med-H | **0.17 ± 0.14** | 0.19 ± 0.04 | 0.20 ± 0.08 | 0.57 ± 0.18 | 0.61 ± 0.53 | 1.36 ± 0.18 |
| | | Low-H | 0.10 ± 0.07 | 0.11 ± 0.02 | 0.30 ± 0.16 | **0.10 ± 0.04** | 11.58 ± 20.05 | 2.04 ± 0.24 |
| | | incomplete | 0.39 ± 0.08 | **0.18 ± 0.02** | 0.90 ± 0.38 | 2.09 ± 2.43 | 5.38 ± 5.85 | 1.87 ± 0.25 |
| | Amateur | High-H | 0.57 ± 0.43 | **0.10 ± 0.01** | 0.12 ± 0.04 | 2.80 ± 1.59 | 0.15 ± 0.13 | 4.69 ± 0.41 |
| | | Med-H | 0.26 ± 0.26 | 0.20 ± 0.06 | **0.11 ± 0.06** | 0.91 ± 0.65 | 0.30 ± 0.22 | 1.42 ± 0.16 |
| | | Low-H | 0.31 ± 0.19 | 0.11 ± 0.03 | **0.09 ± 0.03** | 0.33 ± 0.50 | 0.77 ± 1.05 | 3.24 ± 0.52 |
| | | incomplete | 0.84 ± 0.62 | 0.56 ± 0.09 | **0.34 ± 0.07** | 3.59 ± 1.77 | 2.12 ± 3.27 | 3.02 ± 0.21 |
| MO-Hopper-3obj ($\times 10^5$) | Expert | High-H | 0.19 ± 0.05 | **0.07 ± 0.00** | 0.32 ± 0.03 | 0.26 ± 0.01 | 0.39 ± 0.41 | 0.19 ± 0.10 |
| | | Med-H | 0.18 ± 0.06 | 0.17 ± 0.08 | 0.18 ± 0.03 | 0.23 ± 0.05 | **0.14 ± 0.04** | 0.27 ± 0.08 |
| | | Low-H | 0.19 ± 0.09 | 0.13 ± 0.05 | 0.31 ± 0.17 | **0.05 ± 0.02** | 0.00 ± 0.00 | 1.42 ± 0.37 |
| | | incomplete | 0.17 ± 0.01 | **0.13 ± 0.01** | 0.22 ± 0.11 | 0.26 ± 0.02 | 0.25 ± 0.04 | 0.30 ± 0.09 |
| | Amateur | High-H | 0.32 ± 0.10 | **0.10 ± 0.00** | 0.25 ± 0.09 | 2.41 ± 0.87 | 0.61 ± 0.28 | 0.21 ± 0.12 |
| | | Med-H | 0.25 ± 0.11 | 0.27 ± 0.06 | **0.18 ± 0.04** | 3.74 ± 2.03 | 0.23 ± 0.05 | 0.32 ± 0.09 |
| | | Low-H | 0.34 ± 0.33 | 0.11 ± 0.02 | **0.07 ± 0.03** | 12.17 ± 11.76 | 0.11 ± 0.07 | 1.48 ± 0.47 |
| | | incomplete | 0.28 ± 0.10 | 0.30 ± 0.13 | **0.22 ± 0.07** | 0.78 ± 0.20 | 0.34 ± 0.15 | 0.37 ± 0.13 |
| MO-Swimmer ($\times 10^0$) | Expert | High-H | 4.17 ± 1.27 | 4.43 ± 0.38 | 6.76 ± 2.14 | 6.43 ± 3.98 | 13.36 ± 8.70 | **1.28 ± 0.26** |
| | | Med-H | 3.80 ± 1.12 | 4.26 ± 0.32 | 3.87 ± 0.62 | 5.58 ± 1.70 | 22.07 ± 22.94 | **1.02 ± 0.14** |
| | | Low-H | 31.26 ± 25.30 | 11.36 ± 3.12 | 6.20 ± 2.92 | 13.19 ± 14.24 | 4.77 ± 2.70 | **3.62 ± 0.32** |
| | | incomplete | 5.68 ± 0.70 | 5.76 ± 0.45 | 6.51 ± 3.21 | 5.10 ± 1.12 | 110.54 ± 157.85 | **1.59 ± 0.17** |
| | Amateur | High-H | 5.69 ± 0.89 | 9.50 ± 0.59 | 1.27 ± 0.63 | 10.46 ± 17.93 | 1.50 ± 0.06 | **1.24 ± 0.48** |
| | | Med-H | 4.73 ± 1.10 | 3.68 ± 1.02 | 1.64 ± 0.61 | 2.47 ± 1.99 | 1.44 ± 0.86 | **1.19 ± 0.29** |
| | | Low-H | 10.28 ± 8.03 | 5.32 ± 1.42 | 9.09 ± 6.48 | 5.76 ± 6.21 | 11.88 ± 15.79 | **3.78 ± 0.59** |
| | | incomplete | 4.84 ± 2.09 | 5.36 ± 1.19 | 1.62 ± 0.82 | 4.83 ± 3.37 | **1.06 ± 0.31** | 1.51 ± 0.28 |
| MO-Walker2d ($\times 10^4$) | Expert | High-H | **0.10 ± 0.01** | 0.11 ± 0.01 | 0.46 ± 0.14 | 0.43 ± 0.10 | 18.93 ± 16.20 | 1.42 ± 0.23 |
| | | Med-H | **0.11 ± 0.01** | 0.14 ± 0.02 | 0.45 ± 0.17 | 0.91 ± 0.14 | 13.49 ± 9.87 | 0.46 ± 0.09 |
| | | Low-H | **0.03 ± 0.00** | 0.07 ± 0.01 | 1.66 ± 2.15 | 0.14 ± 0.13 | 1.35 ± 2.33 | 0.47 ± 0.08 |
| | | incomplete | **0.21 ± 0.04** | 0.29 ± 0.02 | 0.47 ± 0.08 | 0.63 ± 0.25 | 8.68 ± 2.18 | 1.17 ± 0.31 |
| | Amateur | High-H | 0.74 ± 0.52 | 0.25 ± 0.03 | **0.18 ± 0.01** | 9.55 ± 2.09 | 1.64 ± 0.58 | 1.68 ± 0.86 |
| | | Med-H | **0.21 ± 0.15** | 0.26 ± 0.07 | 0.24 ± 0.12 | 3.44 ± 2.07 | 2.86 ± 0.83 | 0.56 ± 0.17 |
| | | Low-H | 0.13 ± 0.06 | **0.08 ± 0.01** | 0.09 ± 0.03 | 12.52 ± 19.53 | 7.00 ± 11.76 | 0.49 ± 0.31 |
| | | incomplete | **0.18 ± 0.02** | 0.20 ± 0.05 | 0.29 ± 0.06 | 0.26 ± 0.32 | 2.07 ± 1.60 | 1.32 ± 0.71 |
| Best Count (total=48) | | | **13** | 8 | 10 | 5 | 5 | 7 |

Table 11: The full results on Return Mismatch metric, on `incomplete High-H` datasets

| Environment | Quality | DIFFMORL | MODULI | MORvS | MODT | BC | CQL |
|---|---|---|---|---|---|---|---|
| MO-Ant ($\times 10^2$) | Expert | **2.10 ± 0.14** | 2.20 ± 0.20 | 2.27 ± 0.50 | 5.62 ± 3.43 | 5.83 ± 0.50 | 8.73 ± 0.37 |
| | Amateur | 3.44 ± 0.21 | 3.21 ± 0.68 | **2.32 ± 0.32** | 33.19 ± 2.76 | 8.53 ± 4.08 | 6.32 ± 0.26 |
| MO-HalfCheetah ($\times 10^2$) | Expert | **1.92 ± 0.31** | 2.32 ± 0.20 | 3.27 ± 0.11 | 3.28 ± 0.09 | 5.01 ± 0.05 | 6.12 ± 0.17 |
| | Amateur | 2.67 ± 0.56 | 2.77 ± 0.36 | **2.15 ± 0.18** | 2.46 ± 0.04 | 5.05 ± 0.55 | 5.98 ± 0.10 |
| MO-Hopper ($\times 10^3$) | Expert | **2.46 ± 0.80** | 2.52 ± 0.36 | 2.73 ± 0.31 | 3.89 ± 0.04 | 5.88 ± 2.65 | 3.67 ± 0.91 |
| | Amateur | **2.09 ± 0.72** | 2.36 ± 0.59 | 2.29 ± 0.45 | 2.84 ± 0.29 | 4.63 ± 2.87 | 3.49 ± 0.82 |
| MO-Hopper-3obj ($\times 10^3$) | Expert | 2.99 ± 0.12 | 2.46 ± 0.19 | 1.93 ± 0.28 | 2.86 ± 0.14 | **1.26 ± 0.40** | 3.73 ± 0.84 |
| | Amateur | 2.53 ± 0.58 | 2.21 ± 0.17 | **1.55 ± 0.64** | 2.08 ± 0.48 | 1.84 ± 0.82 | 3.52 ± 0.14 |
| MO-Swimmer ($\times 10^0$) | Expert | **5.92 ± 2.28** | 6.01 ± 1.74 | 11.21 ± 4.27 | 39.46 ± 19.44 | 48.56 ± 54.26 | 56.06 ± 4.38 |
| | Amateur | **17.91 ± 4.93** | 28.72 ± 5.68 | 39.72 ± 4.27 | 114.63 ± 2.11 | 40.75 ± 1.70 | 42.56 ± 18.91 |
| MO-Walker2d ($\times 10^2$) | Expert | **7.62 ± 1.17** | 8.33 ± 1.36 | 13.45 ± 3.35 | 15.19 ± 5.32 | 26.07 ± 0.83 | 25.93 ± 5.28 |
| | Amateur | **4.64 ± 2.87** | 5.38 ± 2.31 | 6.72 ± 2.03 | 30.03 ± 0.11 | 24.36 ± 6.01 | 20.97 ± 3.74 |
| Best Count (total=12) | | **8** | 0 | 3 | 0 | 1 | 0 |

## D.4 VISUALIZATION OF PARETO FRONTS

To intuitively demonstrate the performance and generalization of different methods, we visualize the Pareto fronts of all methods on all environments and all tasks, as shown in Figure 9 to 13. We

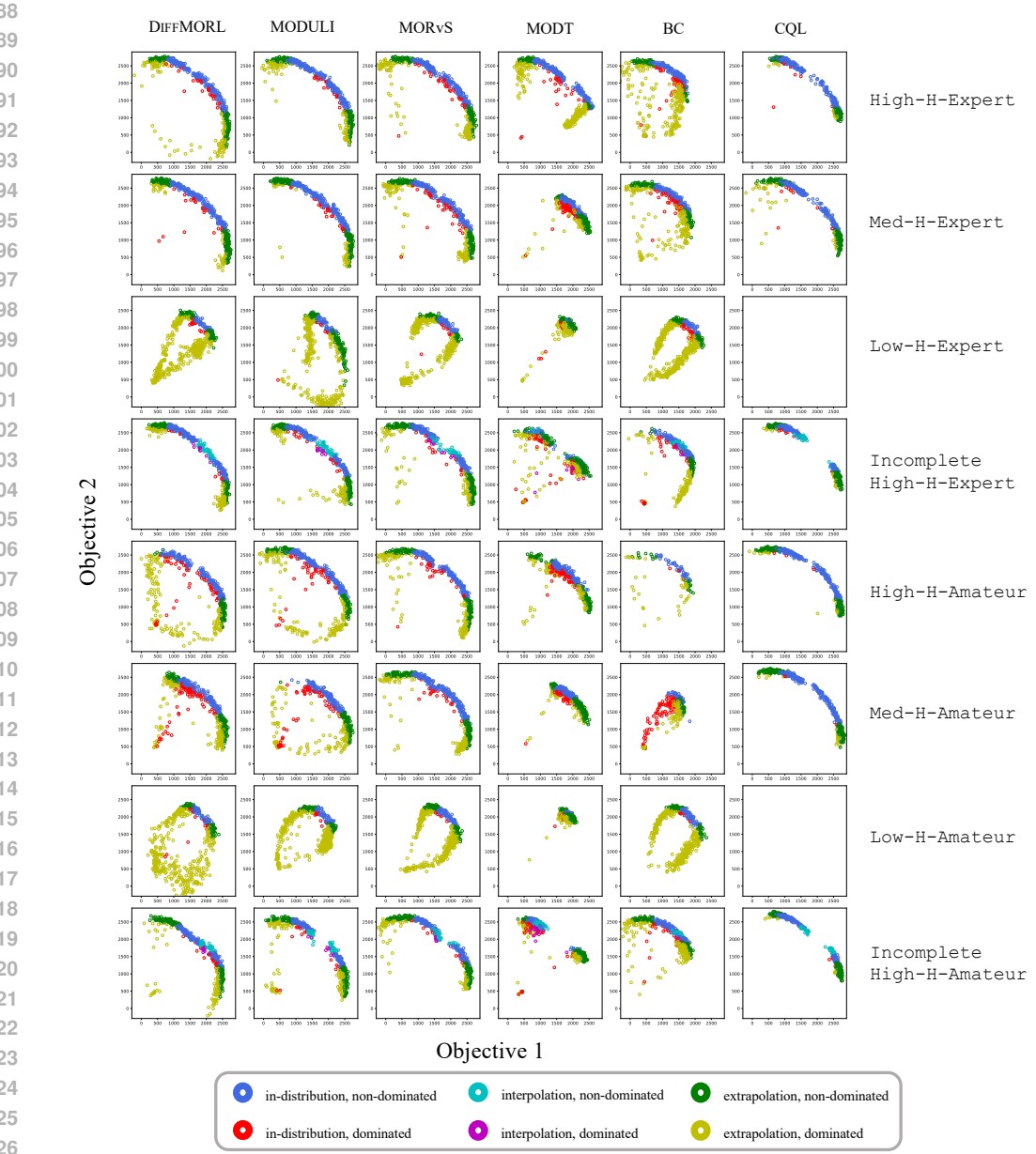

Figure 9: Pareto fronts of different methods on MO-Ant

assign different color for rollouts that correspond to in-distribution, interpolation and extrapolation preference respectively. Overall, we find that our method DIFFMORL, MODULI and MORvS produce significantly better, wider and denser Pareto fronts than MODT, BC and CQL. However, DIFFMORL performs at least comparably well as MODULI and MORvS, and can sometimes outperforms them significantly in more complex tasks such as `Incomplete High-H-Expert` dataset of MO-HalfCheetah, `Low-H-Amateur` dataset of MO-Swimmer, indicating the remarkable performance and generalization ability of DIFFMORL. Note that some Pareto fronts are blank since the corresponding methods cannot produce feasible policies.

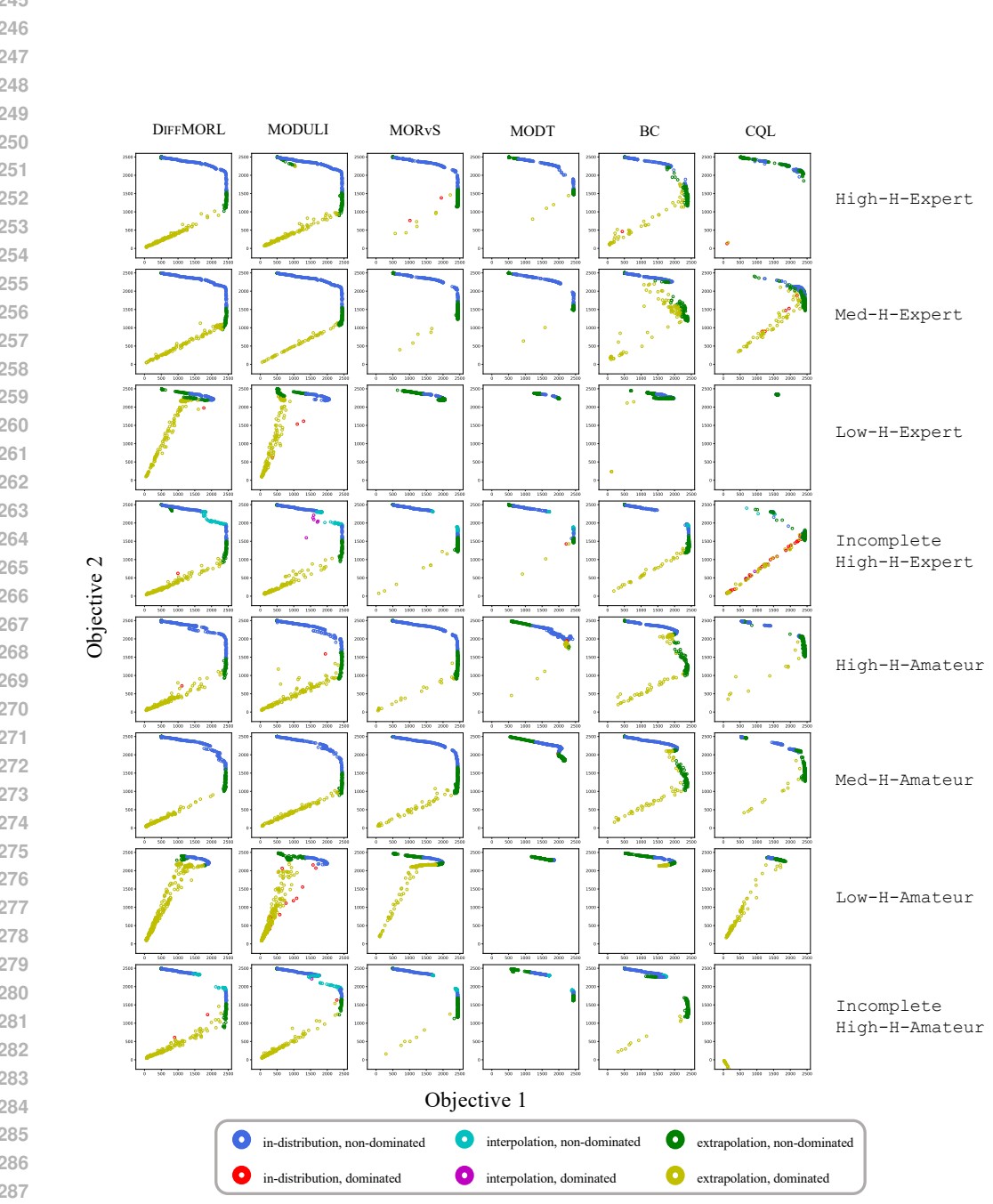

Figure 10: Pareto fronts of different methods on MO-HalfCheetah

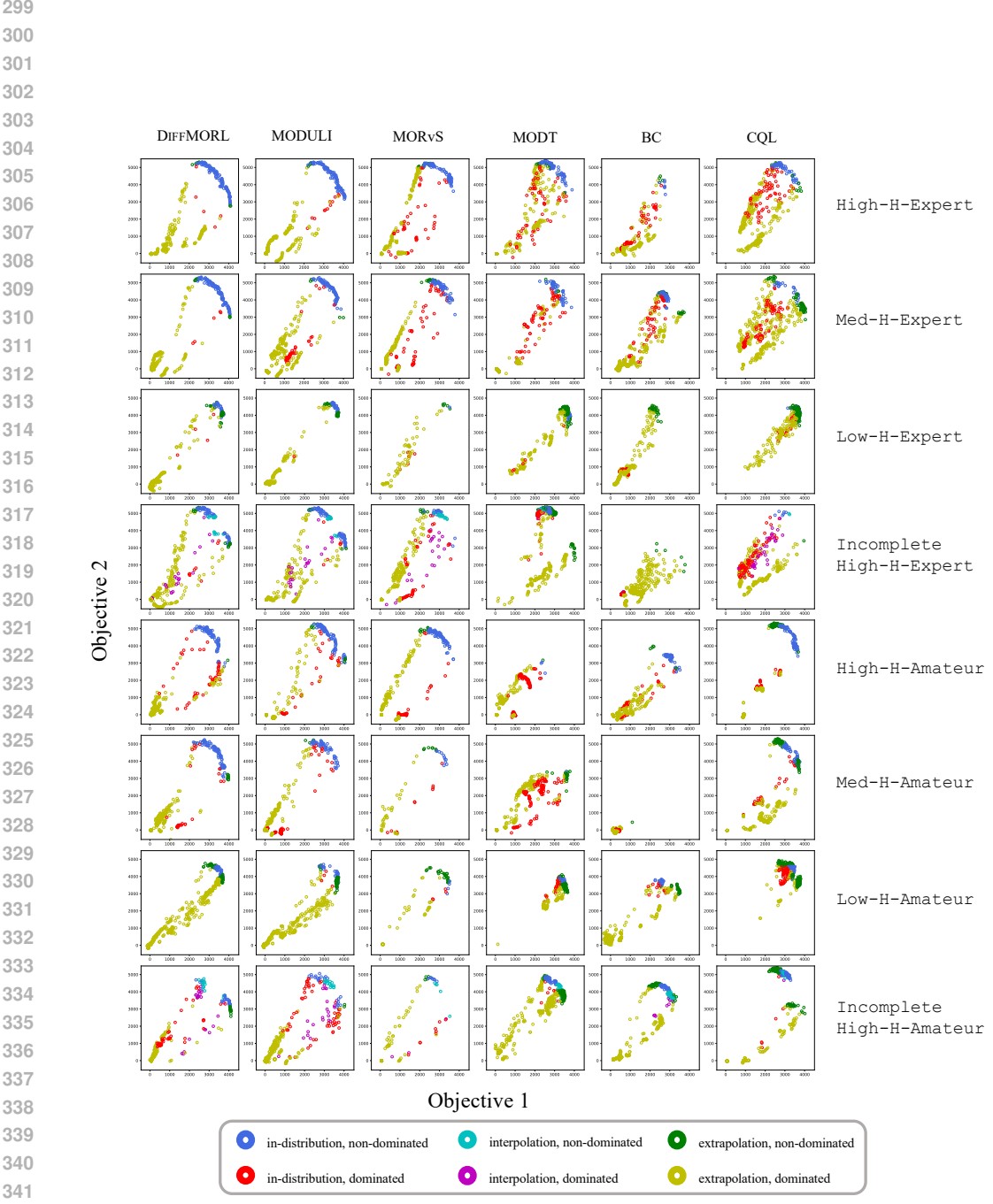

Figure 11: Pareto fronts of different methods on MO-Hopper

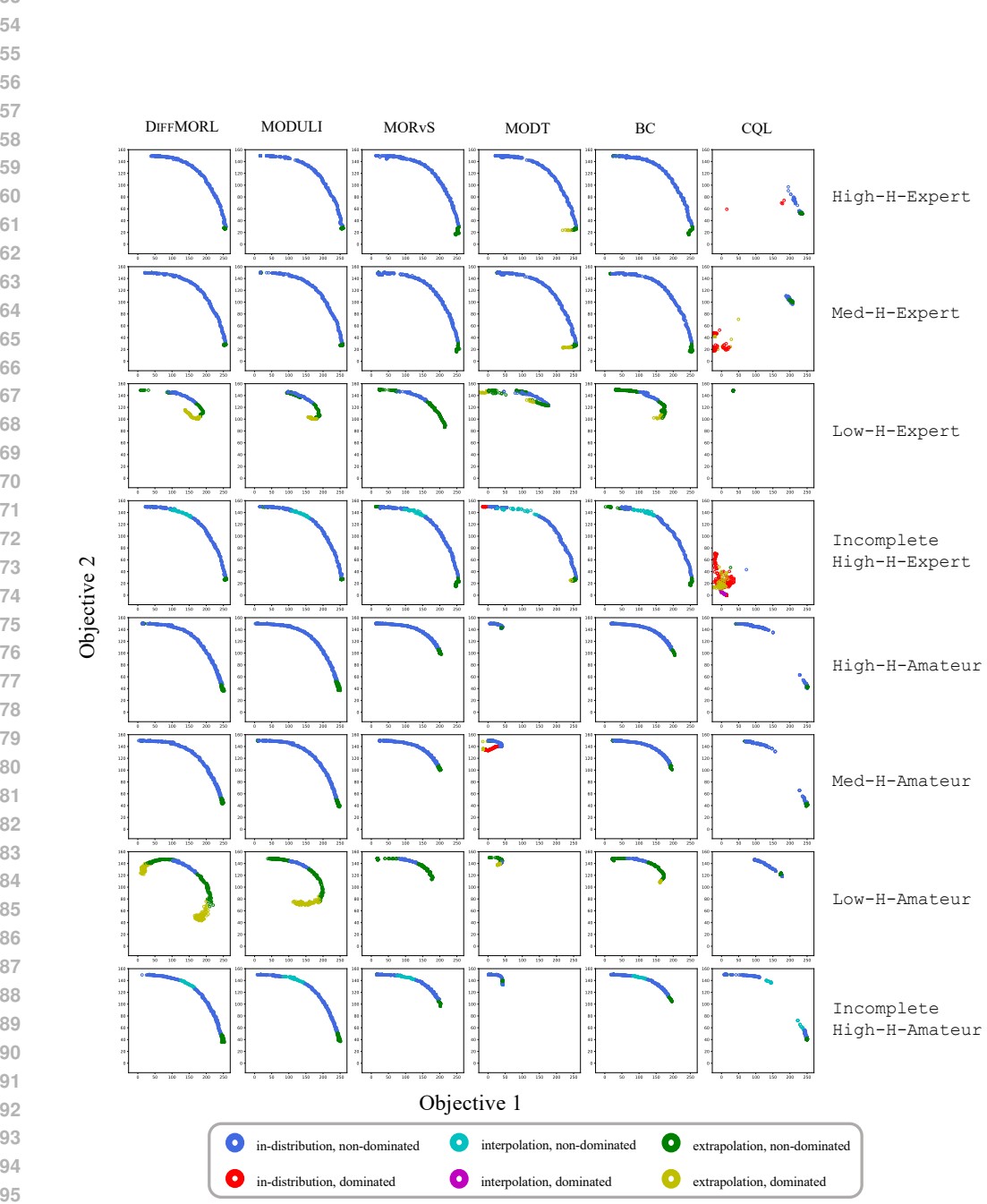

Figure 12: Pareto fronts of different methods on MO-Swimmer

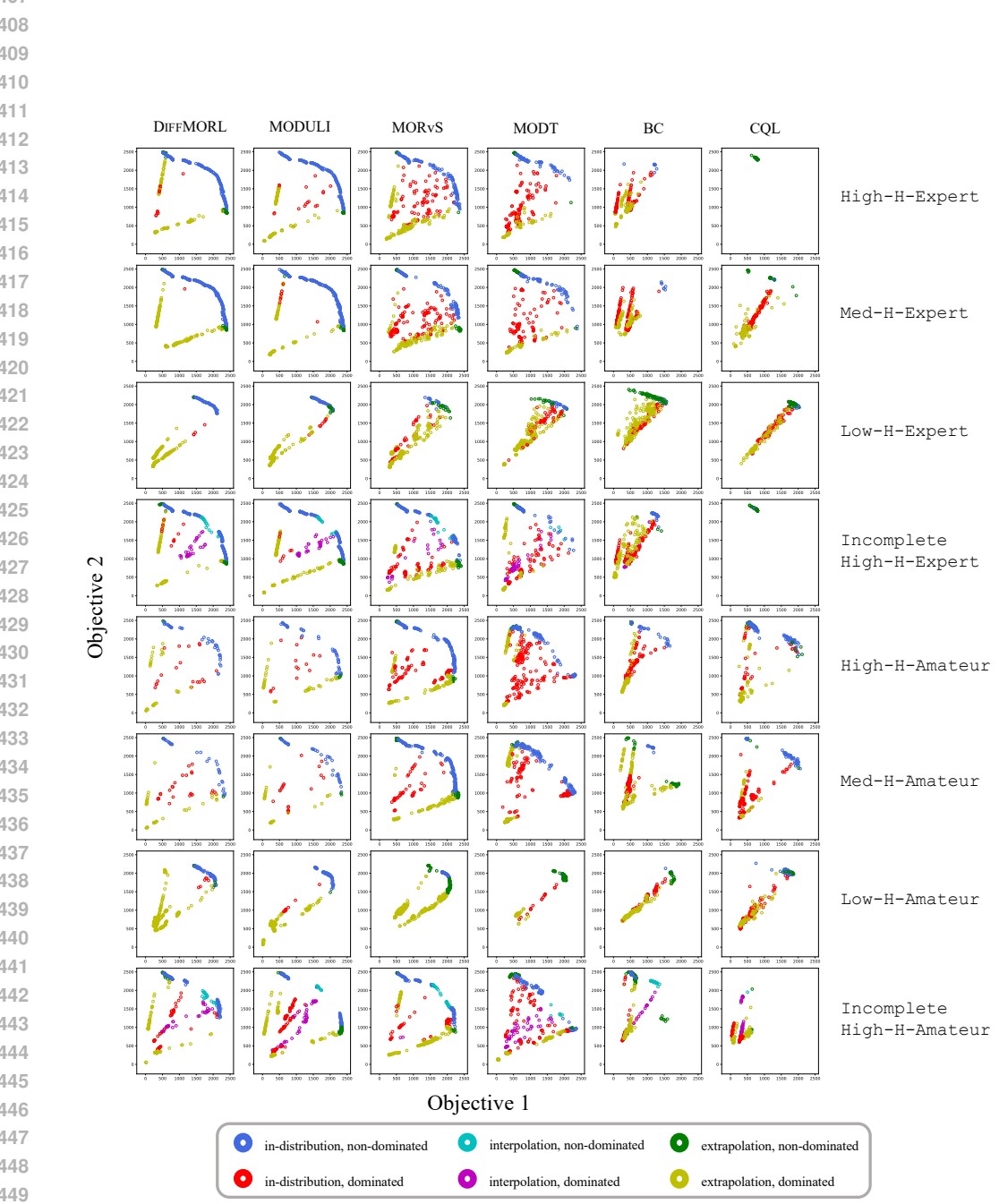

Figure 13: Pareto fronts of different methods on MO-Walker2d

