# OpenReview forum: "Boosting Offline Multi-Objective Reinforcement Learning via Preference Conditioned Diffusion Models"
_ICLR.cc/2025/Conference — ICLR 2025 Conference Withdrawn Submission_

### Official Review · Reviewer_yJnA · 2024-10-21

**Soundness:** 2
**Presentation:** 3
**Contribution:** 2
**Rating:** 3
**Confidence:** 5

**Summary:**

This paper proposes using diffusion planning to address offline multi-objective reinforcement learning (MORL) problems. The authors employ linear interpolation to augment the offline dataset to enhance the algorithm’s generalization capability for out-of-distribution (OOD) preferences. Experimental results on the D4MORL benchmark demonstrate the performance advantages of the proposed method, DIFFMORL, and ablation studies further highlight the benefits of the data augmentation strategy.

**Strengths:**

- Utilizing advanced generative models to capture the complex and diverse distribution of trajectories is a promising approach for tackling offline MORL problems. This method offers better generalization potential compared to learning a narrow policy distribution.

- The authors recognize the challenge of handling OOD preferences, which is a significant obstacle in practical offline MORL applications, and propose a Mixup-based data augmentation technique to address this issue.

**Weaknesses:**

Although DIFFMORL shows some performance improvement over baselines on the D4MORL benchmark, I regret to say that this work offers limited insights and insufficient contributions to the community. In my understanding, DIFFMORL is fundamentally a standard diffusion planning algorithm. Apart from the design of the conditions, it does not differ significantly from decision diffuser (DD). Thus, the key focus of the discussion lies in the new components of DIFFMORL: independent preference encoding (IPE) and mixup-based training (MT).

- IPE is not an innovative design; rather, it is a fairly natural one. In most applications of diffusion models, generating conditions are independently encoded and incorporated into the diffusion network. Within RL community, the idea of encoding different conditions to adjust decision-making objectives has already been discussed in prior work like DD. While IPE is not novel, a more in-depth analysis by the authors—showing exactly how IPE and PC affect trajectory generation and ultimately influence performance—could still be a valuable contribution. Unfortunately, the paper only demonstrates that IPE results in higher scores than PC on MO-HalfCheetah.

- I find the use of MT confusing. Unless I have misunderstood the paper, the authors appear to perform simple linear interpolation between two batches of data to generate synthetic samples. While it is reasonable to apply linear interpolation to preferences or rewards, I seriously question whether interpolating between trajectories yields meaningful data—more likely, it would generate incoherent or nonsensical behavior. For example, if one trajectory corresponds to a HalfCheetah running at 2 m/s and another at 10 m/s, a linear interpolation with a 0.5 weight would not produce a 6 m/s run, but rather an erratic and malfunctioning control signal. Furthermore, the paper lacks an in-depth analysis of MT, making it difficult to trust its effectiveness. I also remain uncertain about whether this method can be broadly applicable, as it seems to provide benefits only in DIFFMORL based on the current experiments.

**Questions:**

- See Weaknesses.
- What's the authors' opinion of MORvS? Compared to diffusion-based algorithms (such as DIFFMORL and MODULI) and transformer-based algorithms (like MODT), MORvS only utilizes a lightweight MLP as its neural network backbone and naively concatenates all information as input. Yet, it remains a highly competitive baseline. Although DIFFMORL shows improvements over MORvS, considering DIFFMORL’s significant inference cost (10 hours for 2500 steps) and the large difference in parameter size (UNet vs. a small MLP), is it possible that engineering improvements to MORvS could yield even greater performance gains? To be clear, I am not challenging the authors’ proposed algorithm. I am merely expressing my curiosity and surprise at MORvS’s performance.

---

### Official Review · Reviewer_z8TE · 2024-10-24

**Soundness:** 2
**Presentation:** 2
**Contribution:** 1
**Rating:** 5
**Confidence:** 5

**Summary:**

This paper studies applying diffusion trajectory generation algorithms to solve multi-objective Offline RL tasks. Especially the diffusion policy is conditioned on task preference weight and the expected episode return.  During inference, the expected return is set to a high value such that the policy can generate high-quality action. CFG technique is applied to further enhance performance.

**Strengths:**

1. The paper applies diffusion modeling to MORL tasks, which I believe is somewhat new and unexplored.
2. The reported experimental numbers are good and clearly outperform previous baselines that do not leverage diffusion policy.
3. The mixup technique is leveraged to solve the poor preference coverage problem.
4. Detailed ablation studies are conducted.
5. Code is provided.

**Weaknesses:**

Overall, my main concern is the novelty of the methodology used in this work.

1. To me, the paper is basically a combination of the Diffuser and DT methods and mix-up techniques and applies them to solve MORL tasks. I don't see any theoretical challenges posed by combining these methods and applying them in MORL besides implementation and experimentation. The application is still meaningful, though, but this is somewhat below the bar of top conferences like ICLR.

2. I have concerns about the mixup-based data augmentation methods. It basically linearly interpolates reward, preference weight, as well as state and action information in the dataset. The first two are fine. However, for state and action data, I really don't think it makes much sense to linearly add them up together, especially is we want to generate actual trajectory rather than conducting classification. If this technique is so useful, we should do this when training text2image models. I'm sure the generated images won't make any sense if we do this in pretraining.

3. When referring to CFG technique in Line 166, you should cite the original CFG paper rather than song's.

**Questions:**

above

---

### Official Review · Reviewer_JZqf · 2024-11-04

**Soundness:** 3
**Presentation:** 3
**Contribution:** 2
**Rating:** 3
**Confidence:** 4

**Summary:**

This paper proposes to utilize diffusion models for offline multi-objective reinforcement learning. Through carefully designed diffusion planning framework and offline data mixup during training, the proposed method exhibits impressive generalization ability. Experiments on the D4MORL benchmark demonstrate its superior performance compared with extensive baselines.

**Strengths:**

- This paper is well written and well motivated. The proposed method is clear and easy to understand.
- Empirical results are sufficient. This paper conducts various experiments to investigate the performance of the proposed method, which outperforms baselines.

**Weaknesses:**

- Limited novelty. At least for me, using diffusion models to improve data modeling ability in reinforcement learning is not new. It has been proposed and validated by many papers previously. Additionally, the paper augments offline datasets by data mixup, which is also not an originally proposed trick. Previous papers [1,2] have demonstrated that we can augment offline datasets via diffusion synthesis, and thus, the policy performance can be improved. It is better to discuss this technique and explain why you do not consider using it.
- Experimental results are sufficient but not strong enough. In Table 1, the proposed method performs almost similarly to the previous diffusion baseline MODULI.

**Reference**

[1] Diffusion Model is an Effective Planner and Data Synthesizer for Multi-Task Reinforcement Learning, NeurIPS 2023

[2] Synthetic Experience Replay, NeurIPS 2023

**Questions:**

- What are the differences between multi-objective RL and traditional RL problems? Why not add different rewards together so it can be a scalar reward instead of a vector reward? I think it should be discussed more clearly in the preliminaries or the appendix.
- What are the differences between your method and MODULI? I find related sentences in related work, but you do not discuss the differences and your improvements.
- I am curious about the reason for not using episodic return as a metric, which is more common in the context of RL. Since I am unfamiliar with multi-objective RL, please correct me if I am wrong.

---

### Note · Authors · 2024-12-24

I have read and agree with the venue's withdrawal policy on behalf of myself and my co-authors.